# Transition from the topological to the chaotic in the nonlinear Su–Schrieffer–Heeger model

Kazuki Sone[1,2] ✉, Motohiko Ezawa [2], Zongping Gong[2], Taro Sawada[2], Nobuyuki Yoshioka[2,3,4] & Takahiro Sagawa [2,5]

Recent studies on topological materials are expanding into the nonlinear regime, while the central principle, namely the bulk–edge correspondence, is yet to be elucidated in the strongly nonlinear regime. Here, we reveal that nonlinear topological edge modes can exhibit the transition to spatial chaos by increasing nonlinearity, which can be a universal mechanism of the breakdown of the bulk–edge correspondence. Specifically, we unveil the underlying dynamical system describing the spatial distribution of zero modes and show the emergence of chaos. We also propose the correspondence between the absolute value of the topological invariant and the dimension of the stable manifold under sufficiently weak nonlinearity. Our results provide a general guiding principle to investigate the nonlinear bulk–edge correspondence that can potentially be extended to arbitrary dimensions.

Band topology is responsible for the existence and absence of zero modes localized at the edge of the sample, which is known as the bulk–edge correspondence[1,2]. A typical model realizing the nontrivial band topology is the Su–Schrieffer–Heeger (SSH) model[3], which is a one–dimensional tight–binding model with staggered linear couplings. While band topology has been studied mainly in electronic systems such as the celebrated quantum Hall effect[4–6], recent studies have also revealed the existence of topological edge modes in various fields of physics including photonics[7,8], fluid[9], and cold atoms[10]. Unlike the standard Schrödinger equation, the dynamics of such classical or quantum bosonic systems are often described by nonlinear equations[11–15]. There are also attempts to extend the notion of topology to such nonlinear systems[16–34], which have revealed that nonlinear effects can induce topological phase transitions. Especially, there are several studies on nonlinear SSH models[35–46]. The emergence of nonlinearity–induced topological edge modes depends on the amplitude[36,47,48], based on which one can appropriately define the nonlinear topological invariants[39,41,49]. However, in strongly nonlinear regimes, some studies[40,50] have pointed out the disappearance of edge

modes. Thus, the bulk–edge correspondence in arbitrary strength of nonlinearity remains unelucidated.

In this paper, we reveal that the strong nonlinear effect induces the transition from topological edge modes to spatially chaotic zero modes. Such a chaos transition in zero modes is expected to be a universal mechanism of the breakdown of the bulk–edge correspondence in nonlinear systems. Specifically, we find that the spatial distribution of zero modes is captured by a discrete dynamical system. Focusing on a minimal model of one–dimensional nonlinear topological insulators, we analyze the bifurcation in the corresponding dynamical system and reveal that it shows the period–doubling bifurcation to chaos. The bifurcation point of the period–doubling bifurcation corresponds to the parameter where the bulk–edge correspondence collapses. Concerning the physical meaning of the absolute value of a nonlinear topological invariant, we propose that under sufficiently weak nonlinearity, it corresponds to the dimension of the stable manifold in the dynamical system describing the zero modes. We demonstrate such correspondence in a model with long–range hoppings than the minimal model. Just like the minimal model,

[1]Department of Physics, University of Tsukuba, Tsukuba, Ibaraki 305-8571, Japan. [2]Department of Applied Physics, The University of Tokyo, 7-3-1 Hongo, Bunkyo-ku, Tokyo 113-8656, Japan. [3]Theoretical Quantum Physics Laboratory, RIKEN Cluster for Pioneering Research (CPR), Wako-shi, Saitama 351-0198, Japan. [4]Japan Science and Technology Agency (JST), PRESTO, 4-1-8 Honcho, Kawaguchi, Saitama 332-0012, Japan. [5]Quantum-Phase Electronics Center (QPEC), The University of Tokyo, 7-3-1 Hongo, Bunkyo-ku, Tokyo 113-8656, Japan. ✉e-mail: sone@rhodia.ph.tsukuba.ac.jp

the bulk–edge correspondence of higher nonlinear topological invariants can also be broken by the chaos transition. These results can be potentially extended to arbitrary dimensions and thus provide the guiding principle to elucidate the bulk–edge correspondence and its breakdown in nonlinear topological insulators.

## Results

### Setup

Following some previous studies[30,39,41,49,51], we define the nonlinear eigenvalue problem to extend the topological invariants and edge modes to nonlinear lattice systems. Labeling the sites and internal degrees of freedom by $x$ and $j$, we consider the general dynamics $i\partial_t\Psi_j(x) = f_j(\Psi, x)$ with $f_j$ ($j = 1, \cdots, M$) being nonlinear functions. Here $\Psi$ is the abbreviation for $\{\Psi_j(x)\}_{x,j}$. We also assume the $U(1)$ symmetry, $f_j(e^{i\theta}\Psi) = e^{i\theta}f_j(\Psi)$ ($j = 1, \cdots, M$), and the translation invariance, which also exist in prototypical setups of linear topological insulators[5,6]. By imposing these symmetries, we can clearly identify the wavenumber–space description of the nonlinear system as below.

Corresponding to the nonlinear dynamics, we formulate the nonlinear eigenvalue problem by defining a nonlinear eigenvector $\Psi$ and a nonlinear eigenvalue $E$ as a vector and a scalar satisfying $f_j(\Psi, x) = E\Psi_j(x)$. We note that the nonlinear eigenvector $\Psi$ corresponds to a periodically oscillating steady state $\Psi_j(x; t) = e^{-iEt}\Psi_j(x)$ in the original nonlinear dynamics. Furthermore, assuming the Bloch ansatz $\Psi(x) = e^{ikx}\psi(k)$, one can derive the wavenumber–space description of the nonlinear eigenequation:

$$f_j(\psi, k) = E(k)\psi_j(k). \tag{1}$$

Finally, we define the nonlinear topological invariants by substituting linear eigenvectors with nonlinear ones in the definitions of conventional topological invariants. We note that a previous paper[34] discussed the bulk–edge correspondence in nonlinear eigenvalue problems with respect to eigenfrequencies, which however describe linear dynamics of higher–order differential equations. In contrast, we here focus on the situation in which the dynamics itself is nonlinear.

We next explicitly define the nonlinear winding number characterizing a one–dimensional nonlinear topological insulator with the sublattice symmetry[45,52–54] (see "Methods") and its bulk-edge correspondence. We assume that the nonlinear eigenequation (Eq. (1)) has the form of:

$$E\begin{pmatrix}\psi_A\\\psi_B\end{pmatrix} = \begin{pmatrix}0 & q(\psi, k)\\q^\dagger(\psi, k) & 0\end{pmatrix}\begin{pmatrix}\psi_A\\\psi_B\end{pmatrix}, \tag{2}$$

where $q(\psi, k)$ is a matrix parametrized by the wavenumber $k$ and the nonlinear eigenvector $\psi = (\psi_A, \psi_B)$. In this manuscript, we further focus on the case that $q(\psi, k)$ only depends on the wavenumber and the amplitude of the state $\|\psi_A\|^2 + \|\psi_B\|^2$, $q(\psi, k) = q(\|\psi_A\|^2 + \|\psi_B\|^2, k)$ with $\|\cdot\|$ being the vector norm. As in a previous study[49], we consider special solutions where the amplitude is fixed $\|\psi_A\|^2 + \|\psi_B\|^2 = w$ independently

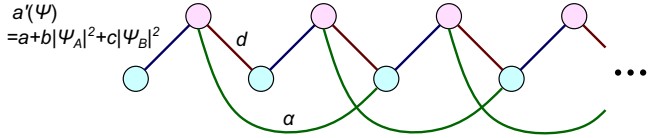

**Fig. 1 | Schematics of the long-range nonlinear SSH model.** The model has two sublattices (red and blue circles) and staggered nearest hopping. The strength of the intercell nearest–neighbor hopping depends on the state variables as $a + b|\Psi_A(x)|^2 + c|\Psi_B(x)|^2$ with $a$, $b$, and $c$ being real parameters. $d$ is also a real parameter that determines the strength of intracell nearest–neighbor hopping. We also introduce long-range hoppings (green curves, the strength $\alpha$) to investigate the effect of higher winding numbers.

of $k$. This assumption is natural because the amplitude is a conserved quantity that one can control by tuning the energy injected to excite the initial state. Then, the nonlinear winding number is defined as:

$$\nu(w) = \frac{1}{2\pi i}\int_0^{2\pi} \partial_k \log(\det q(w, k))dk. \tag{3}$$

The amplitude dependence of the nonlinear winding number implies the potential nonlinearity–induced topological phase transition.

Regarding the bulk–edge correspondence of this nonlinear winding number, we find that the nonzero (zero) winding number basically corresponds to the existence (absence) of the localized zero modes (i.e., nonlinear eigenvectors with zero eigenvalue localized at the edge) when we identify the amplitude $w$ in Eq. (3) to be the edge amplitude, $\sum_j|\Psi_j(1)|^2 = w$. We here assume the zero eigenvalue of topological edge modes because of the sublattice symmetry of the system. Unlike linear systems, one can obtain zero modes even in trivial phases, while they are anti-localized. More specifically, in semi–infinite systems, we fix the edge amplitude $\|\Psi(x = 1)\|^2$ of the zero mode to be $w$. Then, the zero mode should exhibit localization (resp. anti–localization) in the case of $\nu(w) \neq 0$ (resp. $\nu(w) = 0$). However, such a bulk–edge correspondence can be broken by the transition to chaos as we discuss later.

Before moving to the breakdown of the bulk–edge correspondence, we discuss the triviality of the anti–localized zero modes in more detail. First, we can define a topologically trivial phase in nonlinear systems as a phase that can be adiabatically deformed into a linear trivial phase. We can show the existence of such an adiabatic deformation of the phase with the anti–localized zero mode into a linear trivial phase from the fact that one can deform the anti–localized zero mode into a diverging mode by continuously eliminating the nonlinear terms. Since such diverging modes are not regarded as physically feasible edge modes, the divergence indicates the absence of zero modes after adiabatically deforming the nonlinear system into a linear system. Therefore, the anti-localized zero modes are regarded as topologically trivial modes.

### Nonlinear SSH model

To investigate the nonlinear effects on topological edge modes, we consider a nonlinear SSH model, which is a minimal model of one-dimensional nonlinear topological insulators. While previous studies have proposed various types of nonlinear extensions of the SSH model[35–46], we consider one of its variants that is suitable to demonstrate the chaos transition. Specifically, the nonlinear SSH model considered here has two sublattices labeled A and B (cf. Fig. 1), and its dynamics is described as:

$$i\partial_t\Psi_A(x) = (a + b|\Psi_A(x)|^2 + c|\Psi_B(x)|^2)\Psi_B(x) + d\Psi_B(x-1), \tag{4}$$

$$i\partial_t\Psi_B(x) = (a + b|\Psi_A(x)|^2 + c|\Psi_B(x)|^2)\Psi_A(x) + d\Psi_A(x+1), \tag{5}$$

where $\Psi_{A(B)}(x)$ represents the state variables at the A (B) sublattice of the $x$th unit cell. In the following, we focus on the case that $a$, $b$, $c$, and $d$ are real, and $b$ is equal to $c$. We here consider an off-diagonal nonlinearity because it preserves the sublattice symmetry and realizes the nonlinearity–induced topological phase transition[36,49]. One can experimentally realize such an off-diagonal nonlinearity by utilizing, e.g., phase shifts by the Kerr nonlinearity in optical fibers[55] or the nonlinear circuit elements[41] (cf. Supplementary Note 1).

To calculate the nonlinear topological invariant (Eq. (3)), we derive the wavenumber–space description of the nonlinear eigenvalue problem by assuming the Bloch ansatz. The wavenumber–space

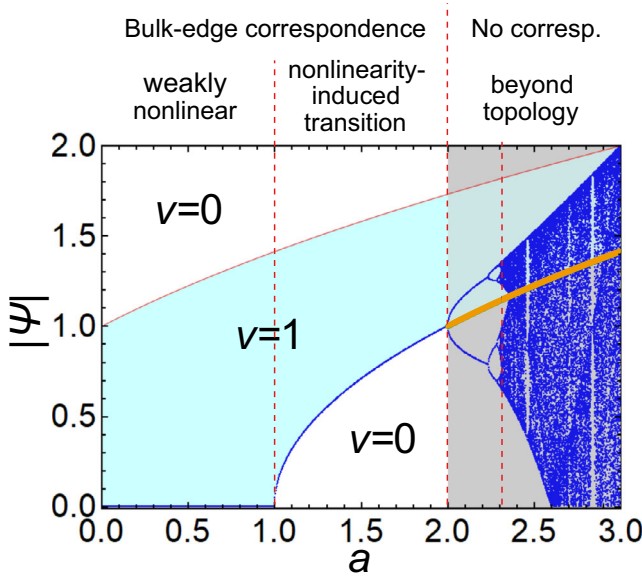

**Fig. 2 | Bifurcation diagram of zero modes in the nonlinear SSH model.** The horizontal axis represents the parameter $a$ determining the strength of non-linearity. The blue curves and dots represent the absolute values of zero modes $|\Psi|$ at each lattice site far from the open boundary. The red dashed lines separate the regions of weakly nonlinear topological phase, nonlinearity-induced topological phase, periodic zero mode phase, and chaotic zero mode phase. As shown in a previous paper[49] for two-dimensional systems, the bulk−edge correspondence is valid from the weak to the mildly strong nonlinear regimes, by introducing the nonlinear extension of the topological number. However, in the truly strong nonlinear regime, the bulk−edge correspondence is broken down by the bifurcation and the subsequent chaos transition. The boundaries at $a = 1, 2$ are analytically obtained from the linear stability analysis of the spatial dynamics, and the boundary around $a = 2.31$ is numerically estimated from the Lyapunov exponent. The light blue area corresponds to the parameter region where the nonlinear winding number $\nu$ becomes one, whose boundaries are analytically obtained. The upper bound of the light blue area (the red curves) corresponds to the unstable fixed points at $|\Psi_A(x)| = \sqrt{a+1}$. The orange curve shows the lower bound of the light blue area ($|\Psi_A(x)| = \sqrt{a-1}$) at $a > 2$, while it does not correspond to the convergent value of the dynamical system, which indicates the breakdown of the bulk-boundary correspondence. The bulk-edge correspondence holds only at $a < 2$, where steady solutions are obtained. The parameters used are $b = c = -1$, $d = 1$. We set the initial condition $\Psi_A(0) = 0.1$.

description of the nonlinear SSH model has the form of Eq. (2):

$$E \begin{pmatrix} \psi_A \\ \psi_B \end{pmatrix} = \begin{pmatrix} 0 & \tilde{a}(\psi) + d e^{-ik} \\ \tilde{a}(\psi) + d e^{ik} & 0 \end{pmatrix} \begin{pmatrix} \psi_A \\ \psi_B \end{pmatrix}, \quad (6)$$

where $\tilde{a}(\psi)$ is a function of $\psi_A$ and $\psi_B$ defined as $\tilde{a}(\psi) = a + b(|\psi_A|^2 + |\psi_B|^2)$. Then, if we focus on the special solutions of nonlinear eigenvectors where $|\psi_A(k)|^2 + |\psi_B(k)|^2 = w$ is fixed independently of the wavenumber $k$, the nonlinear winding number becomes $\nu = \int_0^{2\pi} dk \partial_k \log(a + bw + d e^{ik})/(2\pi i)$. This nonlinear winding number becomes $\nu = 1$ (resp. $\nu = 0$) in the case of $a + bw < d$ (resp. $a + bw > d$), which is consistent with the linear limit $b \to 0$.

## Bifurcation and spatially chaotic zero modes

We analyze the zero mode of the nonlinear SSH model and reveal the breakdown of the bulk−edge correspondence between edge modes and the nonlinear winding number. Specifically, we consider the right semi-infinite system which has an open boundary at $x = 1$. Then, the zero mode has zero amplitude on the B sublattice, i.e., $\Psi_B(x) = 0$, and the spatial distribution on the A sublattice is described by the following

nonlinear dynamical system:

$$\Psi_A(x+1) = -\frac{a + b|\Psi_A(x)|^2}{d} \Psi_A(x) = : F(\Psi_A(x)), \quad (7)$$

where $F(\Psi_A(x))$ is the nonlinear function determining the spatial distribution and independent of $\Psi_B(x)$ and $c$. As discussed above, the bulk−edge correspondence implies that the nonzero winding number corresponds to the existence of the localized zero modes, which is defined as the nonlinear eigenvector with zero eigenvalue and exhibiting a larger amplitude at the edge than in the bulk. We here calculate the zero mode from Eq. (7) and confirm its localization by comparing the edge amplitude with the bulk amplitude; we judge that the zero mode is localized if $|\Psi_A(1)| > \lim_{x \to \infty} |\Psi_A(x)|$. On the other hand, the nonlinear dynamical system (Eq. (7)) is known as the cubic map[56,57] (see "Methods") and shows bifurcations to chaos. We find that this bifurcation leads to the breakdown of the bulk−edge correspondence.

We numerically demonstrate the bifurcation in the zero mode of the nonlinear SSH model by using a bifurcation diagram, which represents the behavior of the dynamical system in Eq. (7) after the relaxation to a steady, periodic, or chaotic solution. In more detail, the bifurcation diagram shows the values of $\Psi_A(x)$ at large $x$ for different $a$'s (we consider $9900 < x < 10,000$ and $0 < a < 3$ in Fig. 2). If we obtain only one value at a fixed $a$ in the bifurcation diagram, the dynamical system (Eq. (7)) converges to that value in the limit of $x \to \infty$. Before a chaos transition, the convergence of the dynamical system is destabilized at a critical value of $a$, and we obtain stable periodic solutions. Such periodic solutions are represented by two or more curves in the bifurcation diagram. After a chaos transition, one obtains scattered points at a fixed $a$, which indicates the unstable and unpredictable dynamics of chaos.

Figure 2 shows the bifurcation diagram of Eq. (7) at $b = -1$, $d = 1$ (see "Methods" for the numerical method). At $a < 2$, we confirm the convergence to a steady value. In particular, in the case of $1 < a < 2$, the convergent value corresponds to the critical amplitude where the nonlinear winding number is changed. This correspondence is not a coincidence, because the boundary between topological and trivial phases must be a threshold where the localization or anti-localization of a zero mode is switched, and thus a spatially uniform mode must appear at the boundary, which is a steady state of Eq. (7). In more detail, when we fix the absolute value at the boundary as $|\Psi_A(1)|^2 = w$, we obtain the following equation from Eq. (7):

$$\Psi_A(2) = -\frac{a + bw}{d} \Psi_A(1). \quad (8)$$

Then, $|\Psi_A(2)|$ becomes larger (resp. smaller) than $|\Psi_A(1)|$ when the nonlinear winding number is $\nu = 1$ (resp. $\nu = 0$), which indicates the localization (resp. anti-localization) of the zero mode in a local sense. In the linear case ($b = 0$), there is no $\Psi_A$ dependence in Eq. (7), and thus the winding number corresponds to the convergence to zero or the divergence. In nonlinear cases, the $\Psi_A$ dependence in Eq. (7) leads to the change of rates of amplification or attenuation of a zero mode, and thus both topological and trivial zero modes can converge to a fixed value. However, such converging solutions from larger (smaller) $|\Psi_A(1)|$ than the convergent value $\lim_{x \to \infty} |\Psi_A(x)|$ still represent spatially localized (anti-localized) nonlinear eigenvectors. Since one can confirm that the convergent value $\lim_{x \to \infty} |\Psi_A(x)|$ must be larger (resp. smaller) than $|\Psi_A(1)|$ at $\nu = 1$ (resp. $\nu = 0$), the nonlinear winding number predicts the localization or anti-localization of the zero mode in this parameter region, which is the bulk−edge correspondence of the nonlinear winding number. As clearly seen from Eq. (7), if we set the edge amplitude $|\Psi_A(1)|^2$ to be the critical amplitude of the nonlinear winding number, $|\Psi_A(1)|^2 = w_c = -(a + d)/b$, $w_c$ also corresponds to the convergent value $w_c = \lim_{x \to \infty} |\Psi_A(x)|^2$ and we obtain a steady state

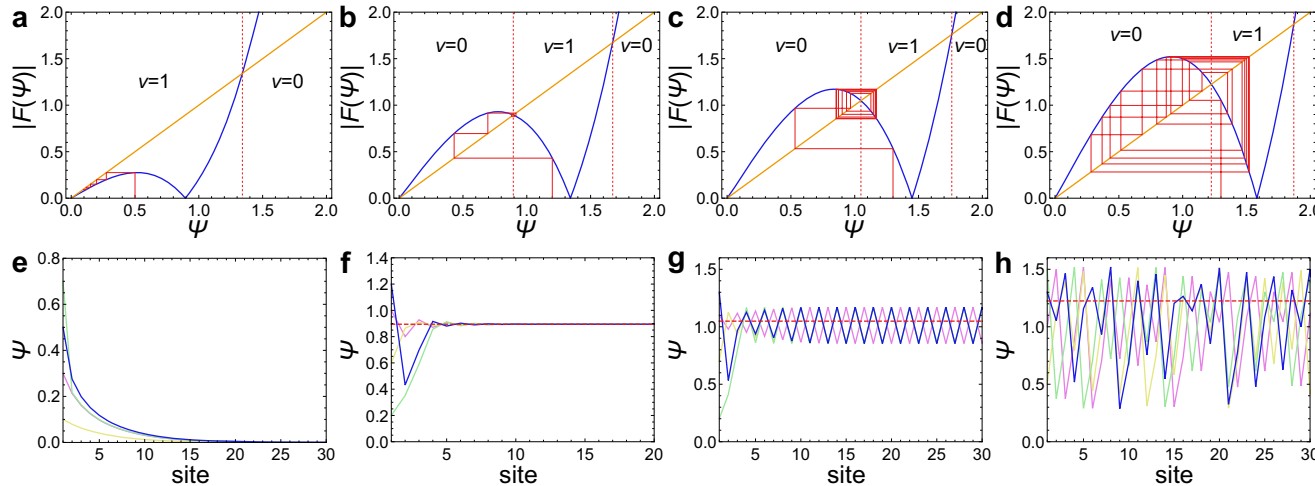

**Fig. 3 | Cobweb plots and spatial distributions of zero modes in the nonlinear SSH model. a–d** Cobweb plots at different strengths of nonlinearity $a$ are shown. The blue curves represent the absolute values of the nonlinear functions $F(\Psi)$ in Eq. (7) with $\Psi$ corresponding to a component of an eigenvector. The orange lines represent $|F(\Psi)| = \Psi$. The red lines show the dynamics of Eq. (7), which correspond to the blue polylines. **a** Weakly nonlinear topological case. If the nonlinear winding number is $\nu = 1$ in the linear limit, localized edge modes converging to zero are obtained for smaller initial amplitudes than that represented by the red dotted line. We use the parameter $a = 0.8$. **b** Nonlinearity-induced topological phase. When the nonlinearity-induced topological phase transition from a trivial phase to a topological phase occurs, localized zero modes are obtained if the initial amplitude is in the region sandwiched by the red dotted lines. We use the parameter $a = 1.8$. **c** Periodic zero mode. We consider $a = 2.1$ and obtain a stable periodic orbit. **d** Chaotic zero mode. At large $a$ ($a = 2.5$ in this panel), we obtain a chaotic dynamics of a zero mode. **e–h** Zero modes at each parameter are shown. The colored polylines show the spatial distributions of zero modes, where the color shows the difference in the initial condition $\Psi_A(1)$. The red dashed lines show the critical amplitude of the nonlinear winding number. We use the parameters $a = 0.8, 1.8, 2.1, 2.5$ in **e–h** for each, which correspond to the upper panels (**a–d**).

where $|\Psi_A(x)|$ is independent of $x$. This is because the critical amplitude of the nonlinear winding number corresponds to the amplitude where the localization or anti-localization of a zero mode is switched.

Meanwhile, at $a > 2$, the nonlinear effect destabilizes the convergence of a zero mode and induces the bifurcation to periodic solutions. Such destabilization of zero modes also breaks their localization, which induces the breakdown of the bulk–boundary correspondence. This period-doubling bifurcation ubiquitously appears before a chaos transition. In fact, we find chaotic zero modes at $a \gtrsim 2.31$ by numerically confirming the positive Lyapunov exponent (see Supplementary Note 2 and Supplementary Fig. 1). Since the convergence values $\lim_{x\to\infty} |\Psi_A(x)|$ are different from the bulk gap-closing points (the orange curve in Fig. 2), one cannot predict such periodic and chaotic zero modes from the bulk topology, which implies the breakdown of the bulk–edge correspondence. We note that while some previous studies[42–44] have discussed the instability and bifurcation in temporal dynamics of nonlinear edge modes, such temporal instability is not regarded as the breakdown of the bulk–boundary correspondence from the viewpoint of nonlinear eigenvalue problems. In contrast, the spatial chaos observed in the present model breaks the localized properties of edge modes, which induces the breakdown of the bulk-boundary correspondence unique to nonlinear systems. Interestingly, however, the temporal instability occurs at the bifurcation point in Fig. 2 and thus may also be related to the spatial instability (see Supplementary Note 3). We also note that since the spatial chaos is only seen in edge modes, such chaos modes are not directly relevant to the statistics of the bulk spectrum.

We also analyze the behavior of zero modes by using cobweb plots, which visualize the nonlinear discrete dynamics in Eq. (7). Figure 3a, e shows the edge modes in the parameter region where the winding number becomes nonzero in the linear limit $b \to 0$. In such a case, we obtain a localized zero mode fully decaying to $\lim_{x\to\infty} \Psi_A(x) = 0$, which corresponds to a conventional topological edge mode. If we consider the case of $1 < a < 2$, the amplitude of the zero mode converges to a nonzero value as shown in Fig. 3b, f. Such a remaining amplitude at $x \to \infty$ is also reported in previous studies[36,41,49] when the $w$-dependent nonlinear

topological invariant signals the nonlinearity-induced topological phase transition. That is, the nonlinear winding number becomes zero (non-zero) at a smaller (larger) amplitude $w$ than the convergent value. While either amplitude gives a converging zero mode, the winding number $\nu(w)$ predicts its localization or anti-localization, which is the nonlinear bulk–edge correspondence. Considering larger $a$, we obtain periodic solutions shown in Fig. 3c, g. In particular, at $x \to \infty$, the zero mode described by the red polyline takes both larger and smaller values than the initial value, and thus one cannot judge the localization or anti-localization of the zero mode. We also confirm the chaotic spatial distribution of the zero mode at $a = 2.5$ (Fig. 3d, h). The spatial chaos of the zero mode is also characterized by the positive Lyapunov exponent $\lambda = 0.728\ldots > 0$ (see also Supplementary Note 2). While we have focused on the semi-infinite system, the bulk–edge correspondence and its breakdown are also confirmed in the finite system of the nonlinear SSH model (see Supplementary Note 4 and Supplementary Fig. 2–4). It is also noteworthy that both the edge modes and the chaos transition are robust against disorders (see Supplementary Note 5 and Supplementary Fig. 5).

We emphasize that the chaos transition of zero modes can be universally seen in a wide range of nonlinear models. At the same time, such a transition is absent in some special models. For example, the SSH model with nonlinearity in the intercell hopping[36,39,41] and the continuum model[49] exhibit no chaos transitions, for which the bulk–boundary correspondence has been confirmed under fairly strong nonlinearity (we show such wide-range stability of edge modes in a nonlinear topological mechanics[46] in Supplementary Note 6 and Supplementary Fig. 6). Thus, the breakdown of the bulk–edge correspondence by chaos transitions has revealed a nontrivial nonlinear effect on topological edge modes.

## Extension to long-range hoppings

In the previous sections, we focus on the model only with nearest-neighbor hoppings and the winding number $\nu(w) = 1$ or 0. Long-range hoppings can lead to nonlinear winding numbers larger than one. In general linear systems, the absolute value of the winding number

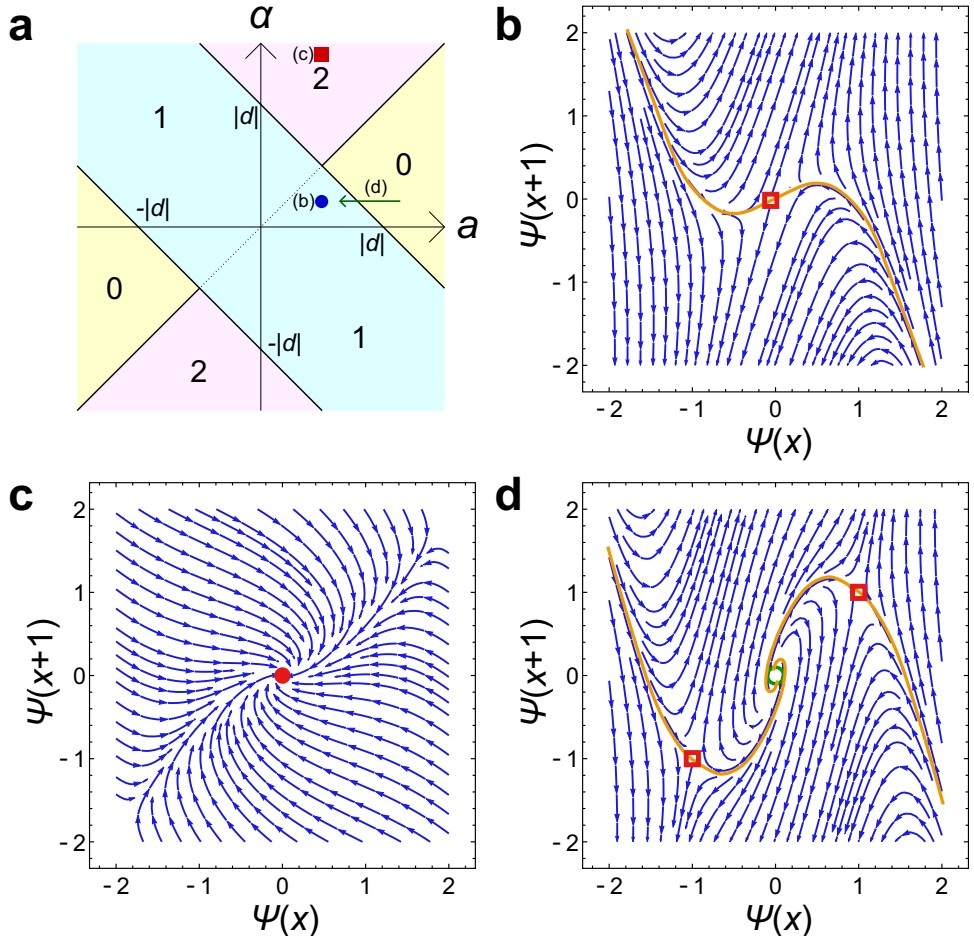

**Fig. 4 | Vector fields representing the deviation of state variables in the extended nonlinear SSH model. a** Phase diagram of the linear extended SSH model. $a$, $d$, and $\alpha$ are real parameters that determine the strength of intercell, nearest-neighbor intracell, and long-range hoppings, respectively. Each color represents the parameter regions with the same winding number, which is shown by the numbers in the regions. **b–d** Vector field and the stable manifold. The blue curved arrows represent the vector field at $(\Psi(x), \Psi(x+1))$ corresponding to the values of eigenvectors at the sites $x$ and $x+1$. The red disks are stable fixed points, the red squares are saddle points, and the green circle is a fully unstable fixed point. The orange curves are eye-guides of the one-dimensional stable manifolds. **b** If the winding number is one under weak nonlinearity (the blue circle in **a**), the stable manifold is one-dimensional. The parameters used are $a = 0.5$, $b = d = -1$, and $\alpha = 0.25$. **c** When the winding number is two under weak nonlinearity (the red filled square in **a**), the fixed point is stable and thus has a two-dimensional stable manifold. The parameters used are $a = 1$, $b = d = -1$, and $\alpha = 4$. **d** When the nonlinearity-induced topological phase transition from zero to one occurs (the green arrow in **a**), nonzero fixed points appear and their stable manifold is one-dimensional. The parameters used are $a = 1.75$, $b = d = -1$, and $\alpha = 0.25$.

corresponds to the number of linearly independent edge modes[58]. Meanwhile, since nonlinear systems have no superposition law, it has been unclear what corresponds to the absolute value of the nonlinear winding number. We here propose that it basically corresponds to the dimension of the stable manifold in the dynamical system describing a zero mode such as Eq. (7). While previous research[44] has also indicated that topological edge modes can be regarded as orbits on stable manifolds, we reveal its correspondence to nonlinear topological invariants.

To analyze the correspondence between the nonlinear winding number and the dimension of the stable manifold, we consider the extended nonlinear SSH model in Fig. 1. In this model, we add next-next-to-nearest-neighbor hoppings ($\alpha\Psi_B(x-2)$ to Eq. (4) and $\alpha\Psi_A(x+2)$ to Eq. (5)) to the nonlinear SSH model. Figure 4a presents the phase diagram of the extended SSH model in the linear limit $b = c = 0$[59–61]. One can confirm that the winding number becomes $\nu = 2$ at $\alpha > d - a$ and $\alpha > a$.

The dynamical system describing the spatial distribution of zero modes in the extended nonlinear SSH model reads:

$$\Psi_A(x+1) = -\frac{a + b|\Psi_A(x-1)|^2}{\alpha}\Psi_A(x-1) - \frac{d}{\alpha}\Psi_A(x), \qquad (9)$$

which looks similar to the Duffing map[62] and the Hénon map[63]. We plot the vector field ($\Psi_A(x+1) - \Psi_A(x)$, $\Psi_A(x+2) - \Psi_A(x+1)$) visualizing the deviations of the state variables at each step of this dynamical system in Fig. 4b–d. When the winding number is one in the linear limit, the fixed point at $\Psi_A = 0$ is a saddle point, and thus its stable manifold is one-dimensional. In contrast, if the winding number is two in the linear limit, the fixed point at $\Psi_A = 0$ is an attractor to which any points in its neighbor converge, and thus the dimension of its stable manifold is two. These results indicate the correspondence between the nonlinear winding number and the dimension of the stable manifold in weakly nonlinear cases. We note that in weakly nonlinear cases, the dimension of the stable manifold corresponds to the number of edge modes in the linear limit, and thus one can expect the correspondence between the winding number and the dimension of the stable manifold for further higher winding numbers.

Under stronger nonlinearity, we find the breakdown of the bulk–edge correspondence in the extended nonlinear SSH model in a sense similar to the bifurcation in the original nonlinear SSH model. If we consider $-(a + \alpha) < d < 0$, $0 < \alpha < a$, $b < 0$, and $(a + \alpha)/|d| < 2$, the model exhibits a nonlinearity-induced topological phase transition and a zero mode converging to $\Psi_A(x) = (a + \alpha + d)/|b|$. In this case, the

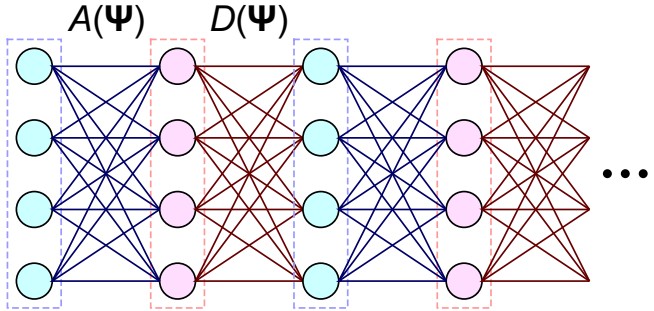

**Fig. 5 | Schematic of a sublattice-symmetric one-dimensional model.** The red and blue filled circles are the sites in this model. The color of each circle represents the sublattice (the red or blue dashed square) to which the site belongs. The lines between the circles show the nonlinear hopping between them. The hopping amplitudes are alternately determined by the matrices $A$ and $D$, which depend on the nonlinear eigenvector $\Psi$.

nonlinear winding number becomes $\nu = 1$ and the dimension of the stable manifold is also one as shown in Fig. 4d, which indicates the bulk-edge correspondence between the nonlinear winding number and the dimension of the stable manifold. However, at $(a + \alpha)/d > 2$, the nonlinear winding number is unchanged, while the fixed point $\Psi_A(x) = (a + \alpha + d)/|b|$ is no longer a saddle point and becomes a fully unstable fixed point (see Supplementary Note 7). At the critical parameter $(a + \alpha)/d = 2$, the bifurcation to a periodic solution occurs. Therefore, the breakdown of the bulk-edge correspondence is induced by the bifurcation as in the original nonlinear SSH model. One can also find that the long-range hopping can increase the number of anti-localized modes corresponding to the higher dimension of the stable manifold (see Supplementary Note 8 and Supplementary Fig. 7).

**More general cases**

While we have focused on the nonlinear SSH model, the bulk-edge correspondence can be extended to more general nonlinear systems. Specifically, we here consider a sublattice-symmetric one-dimensional model whose nonlinear eigenvalue equation is described as:

$$E\begin{pmatrix}\psi_1\\\psi_2\end{pmatrix} = \begin{pmatrix}0 & A^\dagger(g(\psi)) + D^\dagger(g(\psi))e^{-ik}\\A(g(\psi)) + D(g(\psi))e^{ik} & 0\end{pmatrix}\begin{pmatrix}\psi_1\\\psi_2\end{pmatrix},$$
(10)

where $\psi_1$ and $\psi_2$ are $N$-component vectors, and $A(g(\psi))$ and $D(g(\psi))$ are $N \times N$ matrices that are dependent on the nonlinear eigenvectors. We focus on the case that the dependence on the nonlinear eigenvector is determined only by a nonlinear function $g(\psi)$. Figure 5 shows the schematic of this lattice model, where each site belongs to either the first or second sublattice, and the interactions only exist between different sublattices. $A(g(\psi))$ (resp. $D(g(\psi))$) determines the amplitudes of intracell (resp. intercell) couplings. Under the right semi-infinite boundary condition, a nonlinear eigenvector with zero nonlinear eigenvalue follows:

$$\psi_1(x + 1) = T(g(\psi_1(x)))\psi_1(x), \quad \psi_2(x) = 0,$$
(11)

with $T$ being the state-dependent transfer matrix, which is determined by $A(g(\psi))$ and $D(g(\psi))$ as:

$$T(g(\psi_1(x))) = D^{-1}(g(\psi_1(x)))A(g(\psi_1(x))).$$
(12)

Thus, Eq. (11) only depends on the nonlinear function $g(\psi_1(x))$ as in the nonlinear eigenvalue problem (Eq. (10)). Meanwhile, by assuming the Bloch ansatz $\psi(x) = e^{ikx}\psi(k)$ and fixing $g(\psi(k)) = w$ independently of the wavenumber $k$ (in the original nonlinear SSH model, we set

$g(\psi(k)) = \|\psi(k)\|^2)$, one can obtain a nonlinear winding number $\nu(w)$, which also indicates the possible usefulness of fixing nonconservative quantity. Then, if we fix the value of $g(\psi_A(1)) = w$ at the edge of the system, the dynamical system in Eq. (11) at $x = 0$ becomes $\psi_A(2) = T(w)\psi_A(1)$ parametrized by $w$. As is inferred from the linear cases, the nonlinear winding number $\nu(w)$ corresponds to the number of eigenvalues of $T(w)$ whose absolute values are smaller than one (such correspondence can be shown from the argument principle, Supplementary Note 9). Therefore, the correspondence between the nonlinear winding number $\nu(w)$ and amplification or attenuation around the edge site, $\|\psi(1)\| < \|\psi(2)\|$ or $\|\psi(1)\| > \|\psi(2)\|$, can be shown in more general nonlinear systems with arbitrary internal degrees of freedom. If we further assume that the tendency of the amplification or attenuation is unchanged in the limit of $x \to \infty$, one can show the bulk-edge correspondence in such a wide range of nonlinear systems. In contrast, such an assumption is broken by the chaos transition, which can also happen in this general case dependently on $T(w)$. Therefore, the breakdown of the bulk-boundary correspondence by the chaos transition also occurs in a wide range of nonlinear systems. We remain the mathematically rigorous argument as a future work.

While we have focused on sublattice-symmetric systems, topological phases and edge modes can be realized under other symmetries, such as the time-reversal and spatial-inversion symmetries. Under those symmetries, one can use the nonlinear Berry phase as a topological invariant[41] instead of the nonlinear winding number (Eq. (3)). Even without the sublattice symmetry, one can still consider spatial dynamics of edge modes similar to Eq. (7). In such spatial dynamics, it is expected that one can also observe chaos transitions, and thus the breakdown of the bulk-boundary correspondence by chaos transitions universally occurs independently of the symmetries of nonlinear systems.

We also note that there are various nonlinear systems without the $U(1)$ symmetry, such as fluids[13], nonlinear oscillators[14], mechanical lattices[46], and electrical circuits[41]. Such broken $U(1)$ symmetry may alter the topological classification of nonlinear systems, as is the case for interacting bosonic cases. Furthermore, a previous paper[49] has shown that the $U(1)$ symmetry is necessary to obtain the wavenumber-space description of the nonlinear eigenvalue problem (cf. Eq. (2)) from the Bloch ansatz. Possible extension of the topological invariants to one-dimensional systems without $U(1)$ symmetry has been discussed in some previous studies[41,64] by assuming modulated waves instead of the Bloch ansatz. By using a similar technique, we expect that our results of the present paper can be extended to more various systems even without the $U(1)$ symmetry.

## Discussion

We revealed that nonlinear topological edge modes exhibit bifurcations to periodic solutions and chaos by analyzing the dynamical system describing the spatial distribution of zero modes. Such chaos transitions serve as the origin of the breakdown of the bulk-edge correspondence in nonlinear topology. We also proposed that the absolute value of a nonlinear topological invariant corresponds to the dimension of the stable manifold in a dynamical system describing the spatial distribution of zero modes, while such bulk-edge correspondence is also broken by the bifurcation.

While we focused on one-dimensional systems, the analytical techniques used in this paper are applicable to arbitrary dimensions. Specifically, many higher-dimensional topological insulators are reduced to low-dimensional counterparts if we fix the wavenumber to a proper value; for example, the nonlinear Qi-Wu-Zhang model studied in previous research[49] is equivalent to the nonlinear SSH model if we consider the wavenumber in the $y$ direction $k_y = 0, \pi$ (see "Methods"). By using such reduction, one can extend the arguments of chaos in nonlinear topological edge modes to higher-dimensional nonlinear topological insulators. Furthermore, even at $k_y \neq 0, \pi$, one can obtain a nonlinear dynamical system in the spatial direction similar to Eq. (7) (see

also "Methods"). Therefore, the transition to chaos and the breakdown of the bulk-edge correspondence are ubiquitous in nonlinear systems. Meanwhile, our result shows that chaos, a well-known concept in nonlinear dynamical systems, can affect topological physics. Thus, we expect that interplays between nonlinear and topological physics can further uncover the characteristic behaviors in nonlinear topological insulators.

It is noteworthy that $b \neq c$ leads to the difference in the critical amplitudes at which left and right edge modes appear (see Supplementary Notes 10 and 11). Such difference can correspond to a possible $\mathbb{Z} \times \mathbb{Z}$ classification, which is reminiscent of the topological classification of one-dimensional non-Hermitian systems[65–67]. However, since we have considered conserving nonlinear dynamics that correspond to Hermitian systems, the possible $\mathbb{Z} \times \mathbb{Z}$ classification is a genuinely nonlinear effect.

There remains a possibility that one can define other topological invariants beyond the conventional wavenumber-space description and they can recover the bulk-edge correspondence in the chaotic region. Furthermore, while the models analyzed here are conservative dynamics where the sum of the amplitudes are unchanged in the time evolution, there are various dissipative (i.e., non-Hermitian-like) nonlinear systems in nature, such as biological fluids[13] and oscillators[14]. Therefore, the interplay between nonlinear and non-Hermitian topology[68–70] and the extension of the bulk-edge correspondence to dissipative systems remain intriguing future issues.

## Methods
### Sublattice and mirror symmetries of the nonlinear Su–Schrieffer–Heeger (SSH) model
We here propose the definition of the symmetries in nonlinear systems and show that the nonlinear SSH model has sublattice and mirror symmetries. We start from a general nonlinear eigenequation $f_j(\mathbf{\Psi}, x) = E\Psi_j(x)$ (Eq. (1) in the main text). From the analogy to linear systems, we assume that the nonlinear eigenvalues under the sublattice symmetry should accompany their opposite counterparts with the same absolute values and opposite signs. If we consider a map $S$ to such an opposite counterpart, the nonlinear eigenequation should satisfy:

$$f_j(S(\mathbf{\Psi}), x) = -ES(\Psi_j(x)). \quad (13)$$

Then, if the map $S$ is linear to the constant multiple $S(a\mathbf{\Psi}) = aS(\mathbf{\Psi})$, Eq. (13) reads:

$$S^{-1} \circ f_j \circ S(\mathbf{\Psi}, x) = -E\Psi_j(x), \quad (14)$$

where $\circ$ represents the composition of a nonlinear function. Equation (14) is always satisfied when $f_j$ has the symmetry:

$$S^{-1} \circ f_j \circ S = -f_j. \quad (15)$$

Therefore, we adapt this equation as the definition of the sublattice symmetry. We can also define the sublattice symmetries in the wavenumber-space description as:

$$S^{-1} \circ f_j(k) \circ S = -f_j(k). \quad (16)$$

Equations (15) and (16) are derived from each other by assuming the Bloch ansatz.

We can also define the mirror symmetry in a similar way to the sublattice symmetry. Specifically, when the nonlinear eigenequation is described as $f_j(\boldsymbol{\psi}(k), k) = E(k)\psi_j(k)$, the mirror symmetry is defined as:

$$P^{-1} \circ f_j(k) \circ P = f_j(-k), \quad (17)$$

where $P$ is a map representing the mirror operation. The mirror symmetry guarantees that every nonlinear eigenvector appears with its parity-inversion counterpart or is inversion symmetric in itself.

In the nonlinear SSH model, one can find the sublattice and mirror symmetries by considering linear maps $S$ and $P$. Here we denote the right-hand sides of Eq. (6) by $\mathbf{f}(k; (\psi_A, \psi_B))$. By defining $S$ as $S(\boldsymbol{\psi}(k)) = \sigma_z \boldsymbol{\psi}(k)$ with $\sigma_z$ being the $z$-component of the Pauli matrix, one confirms the sublattice symmetry from the following calculation:

$$
\begin{aligned}
&S^{-1}\mathbf{f}(k; S(\psi_A, \psi_B)) \\
&= \sigma_z \mathbf{f}(k; \psi_A, -\psi_B) \\
&= \begin{pmatrix} 1 & 0 \\ 0 & -1 \end{pmatrix} \begin{pmatrix} 0 & a+b|\psi_A|^2 + c|(-\psi_B)|^2 + de^{-ik} \\ a+b|\psi_A|^2 + c|(-\psi_B)|^2 + de^{ik} & 0 \end{pmatrix} \begin{pmatrix} \psi_A \\ -\psi_B \end{pmatrix} \\
&= \begin{pmatrix} 1 & 0 \\ 0 & -1 \end{pmatrix} \begin{pmatrix} -(a+b|\psi_A|^2 + c|\psi_B|^2 + de^{-ik})\psi_2 \\ (a+b|\psi_A|^2 + c|\psi_B|^2 + de^{ik})\psi_1 \end{pmatrix} \\
&= \begin{pmatrix} -(a+b|\psi_A|^2 + c|\psi_B|^2 + de^{-ik})\psi_2 \\ -(a+b|\psi_A|^2 + c|\psi_B|^2 + de^{ik})\psi_1 \end{pmatrix} \\
&= -\mathbf{f}(k; \psi_A, \psi_B).
\end{aligned}
$$
$$(18)$$

Since the nonlinear SSH model holds the sublattice symmetry, one can define the nonlinear winding number. The sublattice symmetry of the nonlinear SSH model also justifies the assumption that the nonlinear eigenvalue of an edge mode is zero.

Similarly, if $b$ is equal to $c$, one can confirm the mirror symmetry of the nonlinear SSH model by defining the mirror operator as $P = \sigma_x$ ($x$-component of the Pauli matrix). In fact, we obtain:

$$
\begin{aligned}
&P^{-1}\mathbf{f}(k; P(\psi_A, \psi_B)) \\
&= \sigma_x \mathbf{f}(k; \psi_B, \psi_A) \\
&= \begin{pmatrix} 0 & 1 \\ 1 & 0 \end{pmatrix} \begin{pmatrix} 0 & a+b|\psi_B|^2 + b|\psi_A|^2 + de^{-ik} \\ a+b|\psi_B|^2 + b|\psi_A|^2 + de^{ik} & 0 \end{pmatrix} \begin{pmatrix} \psi_B \\ \psi_A \end{pmatrix} \\
&= \begin{pmatrix} 0 & 1 \\ 1 & 0 \end{pmatrix} \begin{pmatrix} (a+b|\psi_A|^2 + b|\psi_B|^2 + de^{-ik})\psi_A \\ (a+b|\psi_A|^2 + b|\psi_B|^2 + de^{ik})\psi_B \end{pmatrix} \\
&= \begin{pmatrix} (a+b|\psi_A|^2 + b|\psi_B|^2 + de^{ik})\psi_B \\ (a+b|\psi_A|^2 + b|\psi_B|^2 + de^{-ik})\psi_A \end{pmatrix} \\
&= \mathbf{f}(-k; \psi_A, \psi_B).
\end{aligned}
$$
$$(19)$$

This mirror symmetry is related to the appearance of the gapless point at $k=0$ or $k=\pi$.

### Derivation of the dynamical system describing the spatial distribution of zero modes
We here derive the recurrence relation in Eq. (7) in the main text that describes the spatial distribution of zero modes in the nonlinear SSH model. In the right semi-infinite system, the real-space description of the nonlinear eigenequation becomes as follows:

$$
E \begin{pmatrix} \Psi_A(1) \\ \Psi_B(1) \\ \Psi_A(2) \\ \Psi_B(2) \\ \vdots \\ \vdots \end{pmatrix}
= \begin{pmatrix}
0 & \tilde{a}(\Psi_A(1), \Psi_B(1)) & 0 & 0 & 0 & \cdots \\
\tilde{a}(\Psi_A(1), \Psi_B(1)) & 0 & d & 0 & 0 & \cdots \\
0 & d & 0 & \tilde{a}(\Psi_A(2), \Psi_B(2)) & 0 & \cdots \\
0 & 0 & \tilde{a}(\Psi_A(2), \Psi_B(2)) & 0 & d & \cdots \\
0 & 0 & 0 & d & 0 & \cdots \\
\vdots & \vdots & \vdots & \vdots & \vdots & \ddots
\end{pmatrix}
\begin{pmatrix} \Psi_A(1) \\ \Psi_B(1) \\ \Psi_A(2) \\ \Psi_B(2) \\ \vdots \\ \vdots \end{pmatrix},
$$
$$(20)$$

with $\tilde{a}(\Psi_A(x), \Psi_B(x))$ being the strength of nonlinear hoppings, $\tilde{a}(\Psi_A(x), \Psi_B(x)) = a + b(|\Psi_A(x)|^2 + |\Psi_B(x)|^2)$. To obtain the recurrence relation (Eq. (7)), we assume that the nonlinear eigenvalue is zero, $E = 0$, and $\tilde{a}(\Psi_A(x), \Psi_B(x))$ is nonzero at any $x$. Then, the first row, $\tilde{a}(\Psi_A(1), \Psi_B(1))\Psi_B(1) = 0$ indicates that $\Psi_B(1) = 0$. By iteratively considering the equations in the odd numbers of rows, we obtain $\Psi_B(x) = 0$ at any $x$.

Next, we consider the even numbers of rows. Each of them becomes $\tilde{a}(\Psi_A(x), \Psi_B(x))\Psi_A(x) = d\Psi_A(x+1)$. Therefore, we obtain $\Psi_A(x+1) = \tilde{a}(\Psi_A(x), \Psi_B(x))\Psi_A(x)/d = [a + b(|\Psi_A(x)|^2 + |\Psi_B(x)|^2]\Psi_A(x)/d$, which is equivalent to Eq. (7). Based on the real-space description of the nonlinear eigenequation (Eq. (20)) and its reduction (Eq. (7)), we calculate zero modes of the nonlinear SSH model. We note that this spatial dynamics is different from the temporal dynamics in Eqs. (4) and (5), and we do not calculate zero modes from the steady state of the temporal dynamics.

### Cubic map and its correspondence to the spatial dynamics in Eq. (7)

The cubic map is a nonlinear discrete dynamics defined as:

$$\Phi(t+1) = (1 - a')\Phi(t) + a'\Phi(t)^3, \tag{21}$$

where $\Psi(t)$ is the state variable at time $t$. This discrete dynamics exhibits a chaos transition at $a' \sim 3.3$[56]. One can confirm such a chaos transition from the positivity of the Lyapunov exponent. The nonlinear dynamics in Eq. (7) is equivalent to the cubic map by transforming $a$ and $\Psi(x)$ as $a = (a' - 1)d$ and $\Psi(x) = (-1)^x \sqrt{a'd/b}\Phi(x)$ and assuming the space index $x$ as time $t$. The chaos transition point in Fig. 2 is consistent with the previous study on the cubic map.

### Numerical calculation of the bifurcation diagram of the nonlinear SSH model

To obtain the bifurcation diagram in Fig. 2, we numerically calculate the nonlinear discrete dynamics in Eq. (7). We here fix the parameters $b = -1$ and $d = 1$ and change the parameter $a$. At any $a$, we set the initial condition as $\Psi_A(1) = 0.1$. To obtain $\Psi_A(x)$ at $x > 1$, we iteratively calculate the right-hand side of Eq. (7) and update the value of $\Psi_A(x+1)$ by using $\Psi_A(x)$. We calculate $\Psi_A(x)$ for $1 \le x \le 10000$ at each $a$. Then, we plot the values of $\Psi_A(x)$ of $9900 < x \le 10000$ in Fig. 2. We have also used the same methods in Fig. 3.

### Numerical calculation of the vector field

We plot the vector field $(\Psi_A(x+1) - \Psi_A(x), \Psi_A(x+2) - \Psi_A(x+1))$ in Fig. 4 by using the StreamPlot function in Mathematica.

### Reduction of a higher-dimensional model into a one-dimensional model

As is discussed in the main text, higher-dimensional models of nonlinear topological insulators are often reduced to one-dimensional models, when we consider specific wavenumbers, such as $k_y = 0, \pi$. Therefore, the technique of dynamical systems used in our analysis can be extended to any dimensions, for which our result should provide a guiding principle to investigate the bulk-edge correspondence in nonlinear systems.

We here explicitly show that a two-dimensional model termed the nonlinear Qi-Wu-Zhang (QWZ) model[49] is reduced to the nonlinear SSH model in this article. The nonlinear eigenvalue problem of the nonlinear QWZ model is described in the wavenumber space as:

$$\begin{pmatrix} u + \kappa w(\psi) + \cos k_x + \cos k_y & \sin k_x + i \sin k_y \\ \sin k_x - i \sin k_y & -[u + \kappa w(\psi) + \cos k_x + \cos k_y] \end{pmatrix} \begin{pmatrix} \psi_A(\mathbf{k}) \\ \psi_B(\mathbf{k}) \end{pmatrix} = E \begin{pmatrix} \psi_A(\mathbf{k}) \\ \psi_B(\mathbf{k}) \end{pmatrix}, \tag{22}$$

where $(\psi_A(\mathbf{k}), \psi_B(\mathbf{k}))^T$ is the nonlinear eigenvector at the wavenumber $\mathbf{k} = (k_x, k_y)$, and $w(\psi) = |\psi_A(\mathbf{k})|^2 + |\psi_B(\mathbf{k})|^2$ represents the nonlinear term

in the nonlinear QWZ model. When we focus on the wavenumber $k_y = 0$, the nonlinear eigenvalue problem reads:

$$\begin{pmatrix} u + 1 + \kappa w(\psi) + \cos k_x & \sin k_x \\ \sin k_x & -[u + 1 + \kappa w(\psi) + \cos k_x] \end{pmatrix} \begin{pmatrix} \psi_A(k_x, 0) \\ \psi_B(k_x, 0) \end{pmatrix} = E \begin{pmatrix} \psi_A(k_x, 0) \\ \psi_B(k_x, 0) \end{pmatrix}. \tag{23}$$

Furthermore, if we consider a unitary transformation of the nonlinear eigenvector:

$$\begin{pmatrix} \tilde{\psi}_A(k_x, 0) \\ \tilde{\psi}_B(k_x, 0) \end{pmatrix} = \frac{1}{\sqrt{2}} \begin{pmatrix} \psi_A(k_x, 0) - i\psi_B(k_x, 0) \\ \psi_A(k_x, 0) + i\psi_B(k_x, 0) \end{pmatrix}, \tag{24}$$

Equation (23) becomes:

$$\begin{pmatrix} 0 & u + 1 + \kappa w(\tilde{\psi}) + e^{-ik_x} \\ u + 1 + \kappa w(\tilde{\psi}) + e^{ik_x} & 0 \end{pmatrix} \begin{pmatrix} \tilde{\psi}_A(k_x, 0) \\ \tilde{\psi}_B(k_x, 0) \end{pmatrix} = E \begin{pmatrix} \tilde{\psi}_A(k_x, 0) \\ \tilde{\psi}_B(k_x, 0) \end{pmatrix}, \tag{25}$$

with $w(\tilde{\psi})$ being $w(\tilde{\psi}) = |\tilde{\psi}_A(k_x, 0)|^2 + |\tilde{\psi}_B(k_x, 0)|^2$. Rewriting the parameters $u, \kappa$ as $a = u + 1$, $b = \kappa$, the above eigenequation is equivalent to that of the nonlinear SSH model (Eq. (6)). A similar reduction is conducted at $k_y = \pi$.

For general $k_y$, the unitary transformation in Eq. (24) modifies the nonlinear eigenvalue problem in Eq. (22) as:

$$\begin{pmatrix} \sin k_y & u' + \kappa w(\tilde{\psi}) + e^{-ik_x} \\ u' + \kappa w(\tilde{\psi}) + e^{ik_x} & -\sin k_y \end{pmatrix} \begin{pmatrix} \tilde{\psi}_A(k_x, k_y) \\ \tilde{\psi}_B(k_x, k_y) \end{pmatrix} = E \begin{pmatrix} \tilde{\psi}_A(k_x, k_y) \\ \tilde{\psi}_B(k_x, k_y) \end{pmatrix}, \tag{26}$$

with $u'$ being $u' = u + \cos k_y$. Then, by fixing $k_y$, we obtain an effectively one-dimensional nonlinear eigenequation in the real space as:

$$E\Psi_A(x) = \sin k_y \Psi_A(x) + (u' + \kappa|\Psi_A(x)|^2 + \kappa|\Psi_B(x)|^2)\Psi_B(x) + \Psi_B(x-1), \tag{27}$$

$$E\Psi_B(x) = -\sin k_y \Psi_B(x) + (u' + \kappa|\Psi_A(x)|^2 + \kappa|\Psi_B(x)|^2)\Psi_A(x) + \Psi_A(x+1), \tag{28}$$

which is a nonlinear SSH model with staggered on-site potential. The eigenvalues of nonlinear edge modes in this effectively one-dimensional system are estimated as $E = \sin k_y$ for left-localized modes and $E = -\sin k_y$ for right-localized modes. Assuming such dispersion relation of the edge mode, we obtain a nonlinear dynamical system similar to Eq. (7). Therefore, in the nonlinear QWZ model, the chaos transition of edge modes can occur at arbitrary $k_y$. In general models, specifying the dispersion relation of edge modes can be a nontrivial problem, and thus we have to develop a way to determine the eigenvalue and eigenvector at the same time.

## Data availability
All relevant data to interpret the results of this study are included in the figures. All other data that support the plots within this paper and other findings of this study are available from the corresponding author upon request.

## Code availability
All computational codes used in this study are available from the corresponding author upon request.

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

## Acknowledgements

We thank Hosho Katsura, Eiji Saitoh, and Haruki Watanabe for valuable discussions. K.S. and T. Sawada are supported by World-leading Innovative Graduate Study Program for Materials Research, Information, and Technology (MERIT-WINGS) of the University of Tokyo. K.S. is also supported by JSPS KAKENHI Grant Number JP21J20199. M.E. is supported by JST, CREST Grants Number JPMJCR20T2 and Grants-in-Aid for Scientific Research from MEXT KAKENHI (Grant No. 23H00171). Z.G. is supported by The University of Tokyo Excellent Young Researcher Program. N.Y. is supported by the Japan Science and Technology Agency (JST) PRESTO under Grant No. JPMJPR2119 and JST Grant No. JPMJPF2221. T. Sagawa is supported by JSPS KAKENHI Grant Numbers JP19H05796, JST, CREST Grant Number JPMJCR20C1, and the JST ERATO Grant Number JPMJER2302. N.Y. and T. Sagawa are also supported by Institute of AI and Beyond of the University of Tokyo.

## Author contributions

K.S., M.E., Z.G., T. Sawada, N.Y., and T. Sagawa planned the project. K.S. and M.E. performed the analytical calculations. K.S. also performed the numerical calculations. K.S., M.E., Z.G., T. Sawada, N.Y., and T. Sagawa analyzed and interpreted the results and wrote the manuscript.

## Competing interests

The authors declare no competing interests.
