## [Peer Review File · Nature Communications]

REVIEWER COMMENTS

Reviewer #1 (Remarks to the Author):

The manuscript introduces a nonlinear Su-Schrieffer-Heeger model and studies the bulk-edge correspondence in the weakly and strongly nonlinear regime. They revealed that nonlinear topological edge modes can exhibit the transition to spatial chaos by increasing nonlinearity, and thus establish a universal mechanism of the breakdown of the bulk-edge correspondence. Specifically, they propose a correspondence between the absolute value of the topological invariant and the dimension of the stable manifold under sufficiently weak nonlinearity.

Overall, the manuscript is well-written, and the results are interesting and the calculations are convincing. I believe that the results will be of broad interest to a wide community of researchers working on topology and nonlinear physics. However, I have some technical issues with the criterion for determining the edge state, and consequently its scope of application, which the authors should address. And the comments are listed below.

1. Like the nonlinear dynamical equations of bosonic systems, a derivation of the nonlinear SSH model for Eqs. 4 and 5 is needed.
2. As is mentioned by the authors, the notion of topology has been extended to nonlinear systems and in strongly nonlinear regimes, the bulk-edge correspondence is broken down due to the disappearance of edge modes, thus a clarification between these researches are needed.
3. Anti-Localized state/zero mode has a broad definition than the delocalized one. The authors are responsible to clear the distinction when using.
4. Can chaos transition be judged by spectrum statistics?
5. There are ambiguities in Ψ, Ψ_j and $f(\Psi, x), f(\Psi)$ and other definition.
6. The important findings is that the bifurcation point of the period-doubling bifurcation corresponds to the parameter where the bulk-edge correspondence collapses. What is the physics behind?

7. When the long-range hopping involved, besides the higher nonlinear winding numbers, will it lead to more anti-Localized zero modes comparing to the original nonlinear SSH model?

8. Which Hamiltonian matrix in the article is the bulk Chern number mentioned based on? Is it the one in Eq. 20? Then, according to which equation is the edge state calculated? Is it the eigenstate of the matrix in Eq. 20 at zero energy, or is it the final state evolved from Eqs. 4 and 5 (please explain the criterion for determining whether it is an edge state)?

9. The nonlinear effect normally introduces the dynamical disorder. In the previous research, people found that considering disorder shifts the crossing point of the edge state, namely the position of the Dirac point, which is entirely different from the situation without disorder. In the article, the author still considers the physics at the Dirac point under homogeneous systems. I think that the author needs to explain the reason for this approach and discuss the differences and similarities with the findings of previous studies.

10. More details need for using the cubic map to judge the bifurcation to chaos.

11. The authors need to clarify the behavior of $|\psi\rangle \rightarrow \infty$, non-linear zero mode, and the bifurcation diagram? In Fig. 2, the reason to use $|\psi\rangle \rightarrow \infty$ in the phase diagram?

12. Is the phase boundary in Fig. 2 analytical? If not, how to obtain? What is the topology properties of these boundaries?

13. Stability analysis is important for the nonlinear system, shall they analyze it in Fig. 2?

14. The minimum eigenvalues need to be inversely proportional to the system size and converge to the zero in the thermodynamic limit. For judging the zero mode, one usually needs to check whether the gap closes exponentially. Please check.

Reviewer #2 (Remarks to the Author):

In their manuscript, the authors investigate a nonlinear version of the celebrated SSH model and derive a phase diagram that displays different topological phases, depending on the amplitude of the nonlinearity and a coupling parameter of the linear model. This phase diagram reproduces in a clever way different regimes found in the past (weakly nonlinear and nonlinearity induced transition), but also reveals a transition to chaos which, to my knowledge, is new and interesting.

Coweb plots are also provided, which give a better understanding of the phase diagram. The authors also discuss extended SSH models where long-range hopping terms are considered and are known to yield more edge states and higher winding numbers. Additional studies such as that of the Lyapunov exponents, the stability of the fixed points and a state-dependent transfer matrix approach are carried on, which makes the overall work quite sound.

I find the main result of the paper, namely the transition to chaos and the destruction of the bulk-edge correspondence, very interesting, and I believe this is very nice contribution to the field of nonlinear topology that deserves publication in Nature Communications.

Still, before giving my definitive approval, I would like the authors to clarify a couple of points.

1- The topological analysis is based on a bulk winding number generalized to nonlinear systems and previously introduced, where the amplitude of the nonlinearity is fixed by constraining the norm of Ψ . I understand this as a mathematical trick in order to apply the topological machinery of linear systems to nonlinear ones. However, it seems maybe artificial and also seems to restrict the possible solutions. Could the authors comment about what is captured and what is missed by fixing the norm? Is there a physical justification or is it "just" a technical trick to be able to say something in the very involved investigation of nonlinear topology?

2- The nonlinear SSH model considered here is said to be THE nonlinear SSH model. Obviously, there is not a unique way to introduce nonlinearities in the SSH model. For instance, Eq. (4) and (5) are different from the nonlinear SSH model discussed in section III of Phys. Rev. B 104, 235420 (2021) by one the authors of the present manuscript, where chiral symmetry is broken. The nonlinear SSH model used here actually preserves chiral symmetry, in a sense discussed in Phys. Rev. B 105, 035410 (2022), where again another chiral symmetric nonlinear SSH chain was considered. My question is therefore: how specific is the Hamiltonian considered here? It seems to me that chiral symmetry is a necessary ingredient to have the transition to chaos, otherwise boundary modes are too fragile. Also, chiral symmetry is implicitly used to guarantee that the edge mode lives on one sublattice only, hence Eq.7 from which all the analysis is based on. Could the authors comment on that point?

Reviewer #3 (Remarks to the Author):

In this study, Sone et al. systematically investigate the interplay between nonlinear topological physics and bifurcation. As the triggering amplitude increases, bifurcation leads to chaotic behavior, ultimately resulting in the destruction of nonlinear topological boundary modes. Additionally, the nonlinear topological indices, as demonstrated using the plane-wave ansatz, may break down due to the emergence of chaos. The authors conduct comprehensive analytical and numerical research to explore the relationship between nonlinear topological phases and chaos. Remarkably, the analytical and numerical results align well. Although the study focuses on 1D systems, the findings are readily applicable to higher-dimensional systems.

I find the work to be well-written and intriguing, with the potential to merit publication in the journal Nature Communications. However, I have a couple of suggestions for the authors. If these questions are adequately addressed, I recommend considering this work for publication in Nature Communications.

1. In Eq. (22), the authors specifically consider the special wavenumbers ($k_y = 0$) or (π) to demonstrate the applicability of their results in the 2D system. These wavenumbers serve as special reciprocal-space points where the nonlinear topological modes can be converted to chaotic modes as nonlinearity increases. However, it's essential to explore other wavenumbers as well (i.e., ($k_y \neq 0$) or (π)).

For different wavenumbers, the behavior of nonlinear boundary modes may vary. While the chaotic spatial profile arises for the special wavenumbers, it remains an open question whether similar chaotic behavior occurs for other wavenumbers. Thus, I suggest the authors exemplify the chaotic spatial modes for one or two wavenumbers other than 0 or π .

2. I had a look at the two references, one published in Phys. Rev. Lett. 132, 126601 (2024), "Bulk-Edge Correspondence for Nonlinear Eigenvalue Problems" by Isobe, et. al., the other paper is published by the current authors, namely "Nonlinearity-induced topological phase transition characterized by the nonlinear Chern number", Nature Physics, by Sone et. al. It seems that these two papers have already indicated the nonlinear extension of bulk-boundary correspondence for U(1)-symmetric systems, but now the authors indicate the breakdown of this bulk-boundary correspondence. Is there any inconsistency?

3. In the original ArXiv version of the Nature Physics paper “Nonlinearity-induced topological phase transition characterized by the nonlinear Chern number”, there was a comprehensive discussion on nonlinearity that does not respect $U(1)$ symmetry. However, it appears that this discussion was removed in the published version. Given that many nonlinear interactions in nature violate $U(1)$ symmetry—such as those in mechanical, electrical, and biological systems—it is indeed crucial to address this aspect. I strongly recommend that the authors include a paragraph discussing nonlinear topological physics with interactions that break $U(1)$ symmetry in this current manuscript. This addition would enhance the relevance and completeness of their work.

4. Following the above-mentioned point #3, I recommend that the authors incorporate the reference titled ‘Nonlinear Topological Mechanics in Elliptically Geared Isostatic Metamaterials’ by Ma et al. (Physical Review Letters 131 (4), 046101, 2023). The inclusion of this paper is crucial for the following reason: When studying nonlinear topological mechanical modes in the Kane-Lubensky chain—similar to those investigated in Ref. [39] from the reference list—a chaotic mode should emerge. This behavior arises due to bifurcation in the Kane-Lubensky chain, leading to a chaotic spatial distribution of the mechanical mode. In contrast, the topological mechanical mode presented in ‘Physical Review Letters 131 (4), 046101 (2023)’ does not exhibit a chaotic boundary mode. The absence of chaos in this case results from the lack of bifurcation in the gear rotation angles for the nonlinear topological mechanical boundary mode. A comparative analysis between these two models would be insightful.

5. Following the point #4 raised earlier, I recommend that the authors incorporate the reference titled ‘Topological Boundary Modes in Nonlinear Dynamics with Chiral Symmetry’ by Zhou (arXiv:2403.12480v2). In this work, the authors delve into nonlinear topological physics without $U(1)$ symmetry. It would be insightful to briefly discuss how the breakdown of bulk-boundary correspondence and the emergence of spatially chaotic modes manifest when $U(1)$ -symmetry-breaking nonlinearities are considered.

Overall, I find the analytical derivations to be reasonable, and the numerical results are convincing. Therefore, if these concerns are adequately addressed, I believe it would be appropriate for the manuscript to be published in the journal Nature Communications.

Summary of changes made in the main text

- (1) In response to Reviewer 1's comment 1, we have added the sixth sentence in the first paragraph in the section "Nonlinear SSH model" to discuss the possible realization of the nonlinear SSH model in realistic classical systems. We have also added a new reference [Ref. 55] to refer to a previous paper on topological photonics with nonlinear couplings.
- (2) In response to Reviewer 1's comment 2 and Reviewer 3's comment 2, we have added the third and fourth sentences in the caption of Fig. 2 to clarify the relationship between the bulk-edge correspondence shown in a previous paper and its breakdown discussed in the present manuscript.
- (3) In response to Reviewer 1's comment 3, we have replaced the words delocalization and delocalized with anti-localization and anti-localized to improve the readability.
- (4) In response to Reviewer 1's comment 4, we have added the last sentence in the fourth paragraph in the section "Bifurcation and spatially chaotic zero modes" to clarify the irrelevance of bulk spectrum statistics to the chaos transition of edge modes.
- (5) In response to Reviewer 1's comment 5, we have modified equations (10-12), (18), (19), and the following sentences, and added the second sentence in the third paragraph in the section "Sublattice and mirror symmetries of the nonlinear Su-Schrieffer-Heeger (SSH) model" in Methods to improve the readability.
- (6) In response to Reviewer 1's comment 6, we have modified the ninth sentence in the third paragraph and added the second sentence in the fourth paragraph in the section "Bifurcation and spatially chaotic zero modes" to clarify why the bifurcation point corresponds to the breakdown of the bulk-boundary correspondence.
- (7) In response to Reviewer 1's comment 7, we have added the last sentence in the fourth paragraph in the section "Extension to long-range hoppings" to discuss the increase of the number of anti-localized modes in the long-range SSH model.
- (8) In response to Reviewer 1's comment 8, we have modified and added the first sentence in the second paragraph in the section "Nonlinear SSH model," the fourth and fifth sentences in the first paragraph in the section "Bifurcation and spatially chaotic zero modes," and the fourth and fifth sentences in the second paragraph in the section "Derivation of the dynamical system describing the spatial distribution of zero modes" to clarify the definition of the nonlinear topological invariant and nonlinear edge modes.
- (9) In response to Reviewer 1's comment 9, we have added the last sentences in the third and fourth paragraphs in the section "Sublattice and mirror symmetries of the nonlinear Su-Schrieffer-Heeger (SSH) model" in Methods to clarify that the symmetries in the nonlinear SSH model guarantee the zero eigenvalue of the Dirac point at $k = 0$.
- (10) In response to Reviewer 1's comment 9, we have added the last sentence in the fifth paragraph in the section "Bifurcation and spatially chaotic zero modes" to discuss the robustness of nonlinear edge modes and the chaos transition against spatial disorder.
- (11) In response to Reviewer 1's comment 10, we have modified the fifth sentence in the first paragraph and the third sentence in the fourth paragraph in the section "Bifurcation and spatially chaotic zero modes" and added the section "Cubic map and its correspondence to the spatial dynamics in Eq. (7)" in Methods to clarify the definition of the cubic map and that its chaos transition is judged from the positive Lyapunov exponents.

- (12) In response to Reviewer 1's comment 11, we have modified the first sentence in the fourth paragraph in the section "Setup" and the first and second sentences in the second paragraph in the section "Bifurcation and spatially chaotic zero modes" to clarify the definitions of the words, the behavior of $|\Psi(x)|$ at $x \rightarrow \infty$, nonlinear zero mode, and bifurcation diagram, and to discuss the physical meaning of correspondence between the bifurcation diagram and the phase diagram of nonlinear topological insulators.
- (13) In response to Reviewer 1's comment 12, we have added and modified from the fifth to eighth sentences in the caption of Fig. 2 to clarify how we analytically or numerically obtain the phase boundaries.
- (14) In response to Reviewer 1's comment 13, we have modified and added the sixth and eighth sentences in the fourth paragraph in the section "Bifurcation and spatially chaotic zero modes" to discuss the temporal stability of the nonlinear SSH model and its relation to the bifurcation in Eq. (7).
- (15) In response to Reviewer 2's comment 1, we have modified and added the fifth sentence in the third paragraph in the section "Setup" and the eighth sentence in the first paragraph in the section "More general cases" to discuss the role of fixing amplitude and possibility of fixing other quantities to discuss the bulk-boundary correspondence.
- (16) In response to Reviewer 2's comment 2, we have modified the first sentence and added the second and fifth sentences in the first paragraph in the section "Nonlinear SSH model" to discuss that one can introduce other types of nonlinearities in the SSH model and why we consider off-diagonal nonlinearity.
- (17) In response to Reviewer 2's comment 2, we have added the second sentence in the fourth paragraph in the section "setup" and the last sentence in the third paragraph in the section "Sublattice and mirror symmetries of the nonlinear Su-Schrieffer-Heeger (SSH) model" to clarify the role of the sublattice symmetry in the derivation of Eq. (7). We have also added a new reference [Ref. 45] to refer to a previous work on another type of nonlinear SSH model.
- (18) In response to Reviewer 2's comment 2, we have added the last paragraph in the section "More general cases" to discuss the ubiquity of the chaos transition in nonlinear topological insulators without the chiral symmetry.
- (19) In response to Reviewer 3's comment 1, we have added the third sentence in the second paragraph in the discussion section and the last paragraph in the section "Reduction of a higher-dimensional model into a one-dimensional model" in Methods to discuss the possibility of the chaos transition at nonsymmetric wavenumbers in the nonlinear QWZ model.
- (20) In response to Reviewer 3's comment 2, we have added the last sentences in the second paragraph in the section "Setup" to clarify the difference between the setups considered in a previous work [Ref. 34] and the present paper.
- (21) In response to Reviewer 3's comments 3 and 5, we have added the last paragraph in the section "More general cases" to discuss the importance and possible extensions to nonlinear systems without the $U(1)$ symmetry. We have also added a new reference [Ref. 64] to refer to a recent related work.
- (22) In response to Reviewer 3's comment 4, we have modified the second sentence in the sixth paragraph in the section "Bifurcation and spatially chaotic zero modes" and added a new reference [Ref. 46] to discuss the reason for the absence of the chaos transition in some nonlinear topological insulators.

- (23) We have made minor changes in Fig. 4 and its caption to improve the readability.
- (24) We have added a new citation [Ref. 59] to refer to a previous paper on the linear long-range SSH model that is published earlier than Refs. [59-61] in the previous manuscript. Due to the limitation of the citation, we have removed some references that are less relevant than those added in this revision.
- (25) We have made several minor changes and shortened redundant sentences to improve the readability of the manuscript.

Summary of changes made in the Supplementary Information

- (S1) Related to our change (1) in response to Reviewer 1's comment 1, we have added Supplementary Note 1 to discuss the possibility of the nonlinear coupling by using an optical fiber. We have also added a new citation [Supplementary Ref. 1] to refer to a previous paper that discussed related nonlinear dynamics in optical systems.
- (S2) Related to our change (3) in response to Reviewer 1's comment 3, we have replaced the words delocalization and delocalized with anti-localization and anti-localized to improve the readability.
- (S3) Related to our change (5) in response to Reviewer 1's comment 5, we have modified equations (S1), (S31-S36), and the following sentences to improve the readability. We have also modified the equations in the second and third sentences in the first paragraph in Supplementary Note 7 for the same reason.
- (S4) Related to our change (7) in response to Reviewer 1's comment 7, we have added Supplementary Note 8 to demonstrate the increase in the number of anti-localized modes in the long-range nonlinear SSH model.
- (S5) Related to our change (10) in response to Reviewer 1's comment 9, we have added Supplementary Note 5 to discuss the robustness of edge modes and chaos transitions against spatial disorder. We have also added a new reference [Supplementary Ref. 3] to refer to a previous paper on chaos in disordered systems.
- (S6) Related to our change (13) in response to Reviewer 1's comment 12, we have added the third paragraph and the fourth sentence in the second paragraph in Supplementary Note 2 to clarify that the chaos transition point is estimated from the numerical calculation of the Lyapunov exponents.
- (S7) Related to our change (14) in response to Reviewer 1's comment 13, we have added Supplementary Note 3 to discuss the linear stability in temporal dynamics of the nonlinear SSH model.
- (S8) In response to Reviewer 1's comment 14, we have added from the sixth to tenth sentences in the fourth paragraph in Supplementary Note 4 to discuss the origin of the power decay of the energy gap of nonlinear edge modes.
- (S9) Related to our change (22) in response to Reviewer 3's comment 4, we have added Supplementary Note 6 and a new reference [Supplementary Ref. 5] to discuss the stability of nonlinear edge modes at arbitrary parameters in the topological mechanics discussed in the new reference.
- (S10) We have made several minor changes and shortened redundant sentences to improve the readability of the Supplementary Material.

Reply to Reviewer #1

We are very grateful to Reviewer #1 for their careful reading and the positive appreciation of our work. We have very carefully studied all the comments and criticisms raised by the reviewer. Here, let us reply to each of them.

“The manuscript introduces a nonlinear Su-Schrieffer-Heeger model and studies the bulk-edge correspondence in the weakly and strongly nonlinear regime. They revealed that nonlinear topological edge modes can exhibit the transition to spatial chaos by increasing nonlinearity, and thus establish a universal mechanism of the breakdown of the bulk-edge correspondence. Specifically, they propose a correspondence between the absolute value of the topological invariant and the dimension of the stable manifold under sufficiently weak nonlinearity.

Overall, the manuscript is well-written, and the results are interesting and the calculations are convincing. I believe that the results will be of broad interest to a wide community of researchers working on topology and nonlinear physics. However, I have some technical issues with the criterion for determining the edge state, and consequently its scope of application, which the authors should address. And the comments are listed below.”

We are grateful to the reviewer for the positive appreciation of our results. Below, we address each comment raised by the reviewer.

“1. Like the nonlinear dynamical equations of bosonic systems, a derivation of the nonlinear SSH model for Eqs. 4 and 5 is needed.”

The nonlinear SSH model analyzed in this manuscript is not directly derived from a mean-field analysis of an interacting bosonic system. Nevertheless, we would like to emphasize that this model can describe various systems. Specifically, a previous paper [Ref. 55] has discussed that the Kerr nonlinearity in optical fibers can introduce nonlinear phase shifts in the couplings in an optical lattice. Remaining the leading order terms of such nonlinear phase shifts, one can obtain the nonlinear coupling similar to those in Eqs. (4) and (5). We also mention that nonlinear coupling can be realized by using nonlinear elements in an electrical circuit [Ref. 41], and thus our model may also describe such nonlinear topological electrical circuits. We have added the sentence “One can experimentally realize such an off-diagonal nonlinearity by utilizing, e.g., phase shifts by the Kerr nonlinearity in optical fibers or the nonlinear circuit element (cf. Supplementary Note 1).” in the first paragraph in the section “Nonlinear SSH model,” new references [Ref. 55] and [Supplementary Ref. 1], and Supplementary Note 1 to clarify this point (Summary of changes (1) and (S1)).

“2. As is mentioned by the authors, the notion of topology has been extended to nonlinear systems and in strongly nonlinear regimes, the bulk-edge correspondence is broken down due to the disappearance of edge modes, thus a clarification between these researches are needed.”

In our previous work [Ref. 49], we have extended the notion of topology to nonlinear systems. In that work, we have shown that the notion of band topology is not only applicable to the weakly nonlinear regime but also to the mildly strong nonlinear regime, if we introduce the nonlinear topological number, by which the extended bulk-edge correspondence is restored. In contrast, in the truly strongly nonlinear regime, even the extended bulk-edge correspondence is broken down by bifurcation and chaos transition, which is the very main focus of the present paper. These observations are summarized in Figure 2 in a unified manner, and thus we have added the sentences “As shown in a previous paper for two-dimensional systems, the bulk-edge correspondence is valid from the weak to the mildly strong nonlinear regimes, by introducing the nonlinear extension of the topological number. However, in the truly strong nonlinear regime, the bulk-edge correspondence is broken down by the bifurcation and the subsequent chaos transition.” in its caption for further clarifications (Summary of changes (2)).

“3. Anti-Localized state/zero mode has a broad definition than the delocalized one. The authors are responsible to clear the distinction when using.”

We thank the reviewer for pointing out the necessity of further clarification of the terminologies, anti-localized and delocalized. In contrast to the reviewer’s comment, we used anti-localized in a narrower meaning than delocalized. Anti-localized modes mean nonlinear eigenvectors whose edge amplitude $|\Psi_A(1)|^2 + |\Psi_B(1)|^2$ is smaller than that in the bulk $\lim_{x \rightarrow \infty} (|\Psi_A(x)|^2 + |\Psi_B(x)|^2)$. On the other hand, delocalized modes include both the anti-localized and bulk modes, the latter of which has the same amplitude at the edge and in the bulk. However, we have realized that this terminology was confusing and thus removed all of "delocalization" and only use the terminology "anti-localization" in the revised manuscript (Summary of changes (3) and (S2)).

“4. Can chaos transition be judged by spectrum statistics?”

We consider that the spatial chaos discussed in the present manuscript is not directly relevant to spectrum statistics because the chaos transition is observed in the zero-energy edge mode and thus is independent of the bulk spectrum. We have added the sentence *“We also note that since the spatial chaos is only seen in edge modes, such chaos modes are not directly relevant to the statistics of the bulk spectrum.”* in the fourth paragraph in the section “Bifurcation and spatially chaotic zero modes” to clarify this point (Summary of changes (4)).

“5. There are ambiguities in $\Psi_i, \Psi_{i,j}$ and $f(\Psi_i, x), f(\Psi_i)$ and other definition.”

We thank the reviewer for indicating the necessity of clarification of the definitions of some symbols. Ψ represents the nonlinear eigenvector or the state vector of the nonlinear dynamical system, and Ψ_i is its i -th component. Basically, $f(\Psi)$ and $f(\Psi; x)$ represent the right-hand side of the nonlinear dynamical equation, while we have realized that f has been used to represent other nonlinear functions in some equations (cf. Eqs. (10-12), (18), (19) and Supplementary Eqs. (S1), (S31-S36)). We have modified these equations and the following sentences to clarify this point (Summary of changes (5) and (S3)).

“6. The important findings is that the bifurcation point of the period-doubling bifurcation corresponds to the parameter where the bulk-edge correspondence collapses. What is the physics behind?”

The breakdown of the bulk-edge correspondence at the bifurcation point is induced by the destabilization of the converging solution to the corresponding fixed point. From the viewpoint of the spatial dynamical system, one can regard converging dynamics as topological edge modes because convergence from large $\Psi(x)$ to small $\lim_{x \rightarrow \infty} \Psi(x)$ in spatial dynamics represents the localization of the corresponding nonlinear eigenvector. Then, if such convergence is destabilized and the nonlinear eigenvector oscillates in space, the localization of topological modes can also be broken. We have modified the sentence in the third paragraph to *“However, such converging solutions from larger (smaller) $|\Psi_A(1)|$ than the convergent value $\lim_{x \rightarrow \infty} |\Psi_A(x)|$ still represent spatially localized (anti-localized) nonlinear eigenvectors.”* and added the sentence *“Such destabilization of zero modes also breaks their localization, which induces the breakdown of the bulk-boundary correspondence.”* in the fourth paragraph in the section “Bifurcation and spatially chaotic zero modes” to further clarify this point (Summary of changes (6)).

“7. When the long-range hopping involved, besides the higher nonlinear winding numbers, will it lead to more anti-Localized zero modes comparing to the original nonlinear SSH model?”

As the reviewer anticipates, the long-range hopping can lead to more anti-localized zero modes than the original nonlinear SSH model. We have conducted an additional calculation on the long-range nonlinear SSH model and demonstrate the existence of two anti-localized zero modes in that model (please see Supplementary Note 8 added in the revised manuscript). This emergence of two or more anti-localized zero modes is also understood from the correspondence between the nonlinear topological invariant and the dimension of the stable manifold as in localized zero modes. If we

consider the nonlinearity-induced topological transition at the amplitude w_c where the nonlinear winding number is changed from $\nu = 1$ to $\nu = 2$, there appears a fixed point whose stable manifold is two-dimensional. Corresponding to the converging solutions from $w < w_c$ to that fixed point, we obtain anti-localized modes. Since the stable manifold is two-dimensional, the anti-localized modes have two degrees of freedom in their initial condition, which corresponds to the emergence of two independent anti-localized modes. We have added the sentence “One can also find that the long-range hopping can increase the number of anti-localized modes corresponding to the higher dimension of the stable manifold (see Supplementary Note 8).” in the fourth paragraph in the section “Extension to long-range hoppings” and Supplementary Note 8 to clarify this point (Summary of changes (7) and (S4)).

“8. Which Hamiltonian matrix in the article is the bulk Chern number mentioned based on? Is it the one in Eq. 20? Then, according to which equation is the edge state calculated? Is it the eigenstate of the matrix in Eq. 20 at zero energy, or is it the final state evolved from Eqs. 4 and 5 (please explain the criterion for determining whether it is an edge state)?”

As a matter of fact, we did not calculate the Chern number and instead calculated the nonlinear winding number as a topological invariant, as the SSH model is one-dimensional. The nonlinear winding number is defined by using the wavenumber-space description of the *nonlinear* eigenvalue problem in Eq. (6). Since Eq. (6) falls into a linear eigenvalue problem by assuming the amplitude $|\psi_A(k)|^2 + |\psi_B(k)|^2 = w$ as an additional parameter, we can analytically calculate the nonlinear winding number in Eq. (3). One can also define other nonlinear topological invariants (such as the nonlinear Chern number) by using the nonlinear eigenvalue problem (cf. Ref. [49]).

Second, the edge state is a nonlinear eigenvector of Eq. (20) (the real space expression of the nonlinear eigenvalue problem) with zero energy. More strictly, the edge state is defined as the nonlinear eigenvector that has zero energy and is localized at the edge, whose localization is judged from $|\Psi_A(1)|^2 + |\Psi_B(1)|^2 > \lim_{x \rightarrow \infty} (|\Psi_A(x)|^2 + |\Psi_B(x)|^2)$. Assuming $E = 0$ and $\tilde{a} \neq 0$, this eigenequation falls into Eq. (7), and thus we can calculate the long-range behavior of a zero mode. We emphasize that we do *not* consider the final states of the nonlinear time evolution in Eqs. (4) and (5). We have modified and added the sentence “To calculate the nonlinear topological invariant (Eq. (3)), we derive the wavenumber-space description of the nonlinear eigenvalue problem by assuming the Bloch ansatz” in the second paragraph in the section “Nonlinear SSH model,” the sentences “... which is defined as the nonlinear eigenvector with zero eigenvalue and exhibiting a larger amplitude at the edge than in the bulk. We here calculate the zero mode from Eq. (7) and confirm its localization by comparing the edge amplitude with the bulk amplitude; we judge that the zero mode is localized if $|\Psi_A(1)| > \lim_{x \rightarrow \infty} |\Psi_A(x)|$.” in the first paragraph in the section “Bifurcation and spatially chaotic zero modes,” and the sentences “Based on the real-space description of the nonlinear eigenequation (Eq. (20)) and its reduction (Eq. (7)), we calculate zero modes of the nonlinear SSH model. We note that this spatial dynamics is different from the temporal dynamics in Eqs. (4) and (5), and we do not calculate zero modes from the steady state of the temporal dynamics.” in the second paragraph in the section “Derivation of the dynamical system describing the spatial distribution of zero modes” to clarify these points (Summary of changes (8)).

“9. The nonlinear effect normally introduces the dynamical disorder. In the previous research, people found that considering disorder shifts the crossing point of the edge state, namely the position of the Dirac point, which is entirely different from the situation without disorder. In the article, the author still considers the physics at the Dirac point under homogeneous systems. I think that the author needs to explain the reason for this approach and discuss the differences and similarities with the findings of previous studies.”

As the reviewer points out, if we assume $a + \kappa w = b$, the nonlinear SSH model in the present manuscript has a Dirac point at $k = 0$ and $E = 0$ even under the existence of nonlinearity. This is because the nonlinear term preserves the sublattice and spatial-inversion symmetries (cf. Eqs. (15) and (17)), which the linear model has. The sublattice (resp. spatial inversion) symmetry guarantees that the Dirac point exists at $E = 0$ (resp. $k = 0$). In contrast, the nonlinear Dirac model

investigated in some previous studies (cf. Ref. [51]) breaks the sublattice symmetry and thus can have Dirac points at $E \neq 0$. We have added the sentences in the third and fourth paragraphs “The sublattice symmetry of the nonlinear SSH model also justifies the assumption that the nonlinear eigenvalue of an edge mode is zero.” “This mirror symmetry is related to the appearance of the gapless point at $k = 0$ or $k = \pi$.” in the section “Sublattice and mirror symmetries of the nonlinear Su-Schrieffer-Heeger (SSH) model” in Methods to clarify this point (Summary of changes (9)).

We also find that even under the existence of the spatial disorder that can be induced by nonlinearity, both topological edge modes and the chaos transition remain robust against such inhomogeneity. We have conducted additional calculations of the spatial dynamics of the nonlinear SSH model with the spatially inhomogeneous a , and confirm that the Lyapunov exponent still becomes positive at a certain average of a , which indicates the existence of a chaos transition. We also find that below the bifurcation point, the zero mode almost converges to a critical amplitude, and thus topological edge modes are also robust against disorder. We have added the sentence “It is also noteworthy that both the edge modes and the chaos transition are robust against disorders (see Supplementary Note 5).” in the fifth paragraph in the section “Bifurcation and spatially chaotic zero modes,” Supplementary Note 5, and a new reference [Supplementary Ref. 3] to clarify this point (Summary of changes (10) and (S5)).

“10. More details need for using the cubic map to judge the bifurcation to chaos.”

The cubic map is a discrete nonlinear dynamic defined by $\Phi(t + 1) = (1 - a')\Phi(t) + a'\Phi(t)^3$. The chaos transition in the cubic map is judged from the positive Lyapunov exponents, which we have already calculated in Supplementary Note 2. One can find that the Lyapunov exponent becomes positive around $a' \sim 3.3$, which corresponds to $a \sim 2.3$ in Eq. (7) in the main text, and thus can judge the existence of the chaos transition at this parameter. We have modified the sentence in the fourth paragraph in the section “Bifurcation and spatially chaotic zero modes” to “On the other hand, the nonlinear dynamical system (Eq. (7)) is known as the cubic map (see Methods) and shows bifurcations to chaos.” and added the section “Cubic map and its correspondence to the spatial dynamics in Eq. (7)” in Methods to clarify this point (Summary of changes (11)).

“11. The authors need to clarify the behavior of $|\Psi(x)|_x \rightarrow \infty$, non-linear zero mode, and the bifurcation diagram? In Fig. 2, the reason to use $|\Psi(x)|_x \rightarrow \infty$ in the phase diagram?”

The meanings of the terminologies raised by the reviewer are as follows:

[The behavior of $|\Psi(x)|$ at $x \rightarrow \infty$.] We use this terminology in the second paragraph in the section “Bifurcation and spatially chaotic zero modes” to describe an intuitive meaning of a bifurcation diagram. The detailed meaning of this word is explained in the following sentence as “the values of $\Psi_A(x)$ at large x for different a ’s.” In Fig. 2, we plot the value of $\Psi_A(x)$ at $9900 < x < 10000$ as described in Methods.

[Nonlinear zero mode.] We define the nonlinear zero mode as the nonlinear eigenvector with zero eigenvalue.

[Bifurcation diagram.] The bifurcation diagram is a figure like Fig. 2, which represents the behavior of the nonlinear dynamics after the relaxation to a steady, periodic, or chaotic solution.

While the use of the values of $\Psi_A(x)$ at $x \rightarrow \infty$ in Fig. 2 is the convention in the bifurcation diagram, its relation to the phase diagram has a physical meaning. Specifically, the points in Fig. 2 represent the long-range behavior of localized and anti-localized zero modes at each a . The nonzero data points in Fig. 2 represent the convergence to nonzero values at $x \rightarrow \infty$. This indicates that localization or anti-localization of a zero mode depends on whether the initial amplitude is larger or smaller than the convergent value, which indicates the existence of the nonlinearity-induced topological transition. When periodic or chaotic solutions appear at $x \rightarrow \infty$, we cannot judge the localization or anti-localization of a zero mode from its long-range behavior, which leads to the breakdown of the bulk-boundary correspondence. We have modified the sentence in the fourth paragraph in the section “Setup” to “... we find that the nonzero (zero) winding number basically corresponds to the existence (absence) of the localized zero modes (i.e., nonlinear eigenvectors with

zero eigenvalue localized at the edge) ...” and the sentences in the second paragraph in the section “Bifurcation and spatially chaotic zero modes” to “We numerically demonstrate the bifurcation in the zero mode of the nonlinear SSH model by using a bifurcation diagram, which represents the behavior of the dynamical system in Eq. (7) after the relaxation to a steady, periodic, or chaotic solution. In more detail, the bifurcation diagram shows the values of $\Psi_A(x)$ at large x for different a 's (we consider $9900 < x < 10000$ and $0 < a < 3$ in Fig. 2).” to clarify these points (Summary of changes (12)).

“12. Is the phase boundary in Fig. 2 analytical? If not, how to obtain? What is the topology properties of these boundaries?”

The boundary between the regions of $\nu = 0$ and $\nu = 1$ is analytically obtained. The boundaries at $a = 1$ and $a = 2$ are also analytical, while that around $a = 2.31$ is not exact. Since one can calculate the nonlinear winding number from Eq. (3), the phase boundary of the nonlinear winding number is described as $a + bw = d$ ($w = |\Psi_A(x)|$). Therefore, in Fig. 2, we analytically obtain the boundary between $\nu = 0$ and $\nu = 1$ as $|\Psi_A(x)| = \sqrt{a \pm 1}$. The boundaries at $a = 1$ and $a = 2$ correspond to the transition points of the linear stability of fixed points. We can analytically calculate such transition points from the linear stability analysis of the dynamical system. However, the chaos transition point is difficult to calculate. The boundary around $a = 2.31$ is approximately obtained from the numerical calculation of the Lyapunov exponent in Supplementary Note 2. We have added and modified from the fifth to eighth sentences in the caption of Fig. 2 and added the third paragraph and the sentence “We estimate the chaos transition point in Fig. 2 in the main text from this numerical result.” in the second paragraph in Supplementary Note 2 to clarify these points (Summary of changes (13) and (S6)).

“13. Stability analysis is important for the nonlinear system, shall they analyze it in Fig. 2?”

As is mentioned in the reply to the previous comment, the linear stability of fixed points in Eq. (7) is already conducted and reflected in Fig. 2. Thanks to the reviewer’s comment, however, we also realize that the linear stability of the nonlinear time evolution in Eqs. (4) and (5) is also related to the bifurcation in Fig. 2. Specifically, we consider the linearization of Eqs. (4) and (5) around periodic gapless solutions. Then, the eigenvalues of the matrix obtained by the linearization become complex at $a > 2$, which implies the instability of the zero modes (see Supplementary Note 3 for the details of the calculation). Therefore, the edge modes become both temporally and spatially unstable at the bifurcation point. We have modified and added the sentences “... such temporal instability is not regarded as the breakdown of the bulk-boundary correspondence from the viewpoint of nonlinear eigenvalue problems.” “Interestingly, however, the temporal instability occurs at the bifurcation point in Fig. 2 and thus may also be related to the spatial instability (see Supplementary Note 3).” in the fourth paragraph in the section “Bifurcation and spatially chaotic zero modes” and added Supplementary Note 3 to clarify these points (Summary of changes (14) and (S7)).

“14. The minimum eigenvalues need to be inversely proportional to the system size and converge to the zero in the thermodynamic limit. For judging the zero mode, one usually needs to check whether the gap closes exponentially. Please check.”

Power-law convergence of the nonlinear eigenvalue is a characteristic of nonlinear topological edge modes, which is also confirmed in our previous study [Ref.49]. In linear cases, topological edge modes decay exponentially. Since the interaction between left- and right-edge modes are estimated from their resonance integral, the strength of the interaction also exponentially decreases as the system size increases. In contrast, the nonlinear edge modes can have nonvanishing amplitude even far from the edge. Their resonance integral normalized by their norm is proportional to the inverse of the system size, which is larger than linear cases. Therefore, the gap induced by the finite-size effect does not decay exponentially in the system size. We have added the last several sentences in the fourth paragraph in Supplementary Note 4 to clarify this point (Summary of changes (S8)).

Once again, we are very grateful to Reviewer #1 for the positive appreciation of our work and for the valuable comments. In particular, we have conducted an additional calculation on the temporal stability of the nonlinear SSH model and its relation to the bifurcation in the spatial dynamics. We do hope that the revised manuscript, together with our reply above, will meet with the reviewer's approval.

Reply to Reviewer #2

We are very grateful to Reviewer #2 for their careful reading and the positive appreciation of our manuscript. We have very carefully studied all the comments and criticisms raised by the reviewer. Here, let us reply to each of them.

"In their manuscript, the authors investigate a nonlinear version of the celebrated SSH model and derive a phase diagram that displays different topological phases, depending on the amplitude of the nonlinearity and a coupling parameter of the linear model. This phase diagram reproduces in a clever way different regimes found in the past (weakly nonlinear and nonlinearity induced transition), but also reveals a transition to chaos which, to my knowledge, is new and interesting.

Coweb plots are also provided, which give a better understanding of the phase diagram. The authors also discuss extended SSH models where long-range hopping terms are considered and are known to yield more edge states and higher winding numbers. Additional studies such as that of the Lyapunov exponents, the stability of the fixed points and a state-dependent transfer matrix approach are carried on, which makes the overall work quite sound.

I find the main result of the paper, namely the transition to chaos and the destruction of the bulk-edge correspondence, very interesting, and I believe this is very nice contribution to the field of nonlinear topology that deserves publication in Nature Communications.

Still, before giving my definitive approval, I would like the authors to clarify a couple of points."

We are grateful to the reviewer for the positive appreciation of our results. We address each comment raised by the reviewer below.

"1- The topological analysis is based on a bulk winding number generalized to nonlinear systems and previously introduced, where the amplitude of the nonlinearity is fixed by constraining the norm of Ψ . I understand this as a mathematical trick in order to apply the topological machinery of linear systems to nonlinear ones. However, it seems maybe artificial and also seems to restrict the possible solutions. Could the authors comment about what is captured and what is missed by fixing the norm? Is there a physical justification or is it "just" a technical trick to be able to say something in the very involved investigation of nonlinear topology?"

We consider that the assumption of fixing the amplitude is natural in conservative dynamical systems, because the amplitude is a conserved quantity and proportional to the energy of a nonlinear wave. Therefore, the amplitude is determined from the initial condition and controlled by tuning the energy injected to excite the initial state. We also note that if we sweep amplitude w over all positive values, we can obtain all the eigenvectors described by the Bloch ansatz, because any Bloch ansatz eigenvector is determined by k and w .

On the other hand, as we discussed around Eqs. (10-12), one could fix other quantities (cf. $g(\boldsymbol{\psi}(\mathbf{k}))$ in Eq. (10)) to define the nonlinear topological invariant, if the nonlinear model has more general terms than the Kerr nonlinear term. We have modified and added the sentence "This assumption is natural because the amplitude is a conserved quantity that one can control by tuning the energy injected to excite the initial state." in the third paragraph in the section "Setup" and the sentence "... one can obtain a nonlinear winding number $\nu(w)$, which also indicates the possible

usefulness of fixing nonconservative quantity.” in the first paragraph in the section “More general cases” to clarify these points (Summary of changes (15)).

“2- The nonlinear SSH model considered here is said to be THE nonlinear SSH model. Obviously, there is not a unique way to introduce nonlinearities in the SSH model. For instance, Eq. (4) and (5) are different from the nonlinear SSH model discussed in section III of Phys. Rev. B 104, 235420 (2021) by one the authors of the present manuscript, where chiral symmetry is broken. The nonlinear SSH model used here actually preserves chiral symmetry, in a sense discussed in Phys. Rev. B 105, 035410 (2022), where again another chiral symmetric nonlinear SSH chain was considered. My question is therefore: how specific is the Hamiltonian considered here? It seems to me that chiral symmetry is a necessary ingredient to have the transition to chaos, otherwise boundary modes are too fragile. Also, chiral symmetry is implicitly used to guarantee that the edge mode lives on one sublattice only, hence Eq.7 from which all the analysis is based on. Could the authors comment on that point?”

We thank the reviewer for introducing a relevant paper, which we have cited in the revised manuscript as Ref.45. As the reviewer points out, the nonlinear SSH model analyzed in the present paper is one of the possible ways to introduce nonlinearity into the SSH models. We have modified the sentence “... we consider a nonlinear SSH model ...” and added the sentences “While previous studies have proposed various types of nonlinear extensions of the SSH model, we consider one of its variants that is suitable to demonstrate the chaos transition.” “We here consider an off-diagonal nonlinearity because it preserves the sublattice symmetry and realizes the nonlinearity-induced topological phase transition.” in the first paragraph in the section “Nonlinear SSH model” to clarify this (Summary of changes (16)).

On the other hand, we consider that the chiral symmetry is not necessary for chaos transition in nonlinear topological insulators. The nonlinear SSH models in some previous papers (cf. Refs. [40,41,45]) preserve the time-reversal and spatial-inversion symmetries, under which the nonlinear Berry phase is used as a topological invariant. In such models, the nonlinear dynamical system in the spatial direction becomes complicated because one cannot assume the zero eigenvalue of edge modes and thus both sublattices can have nonzero amplitudes in topological modes. However, one can still consider spatial dynamical systems and relate the convergence to the fixed point with the existence of topological modes. Then, destabilization of such convergent solutions leads to chaos transition, which is also regarded as the breakdown of the bulk-boundary correspondence.

Meanwhile, the chiral symmetry guarantees that the edge mode has nonzero amplitude only on one sublattice and thus we can obtain one-component nonlinear dynamics in the spatial direction in Eq. (7). We have realized that such a role of the chiral symmetry in the derivation of Eq. (7) was not clear in the previous manuscript and thus have added the sentence “We here assume the zero eigenvalue of topological edge modes because of the sublattice symmetry of the system.” in the fourth paragraph in the section “setup” and the sentence “The sublattice symmetry of the nonlinear SSH model also justifies the assumption that the nonlinear eigenvalue of an edge mode is zero.” in the third paragraph in the section “Sublattice and mirror symmetries of the nonlinear Su-Schrieffer-Heeger (SSH) model” and added a new reference [Ref. 45] to clarify this point (Summary of changes (17)). We have also added the last paragraph in the section “More general cases” to discuss the ubiquity of the chaos transition in nonlinear topological insulators without the chiral symmetry (Summary of changes (18)).

Once again, we are very grateful to Reviewer #2 for the positive appreciation of our work. We do hope that the revised manuscript, together with our reply above, will meet with the reviewer’s approval.

Reply to Reviewer #3

We are very grateful to Reviewer #3 for their careful reading and the positive appreciation of our manuscript. We have very carefully studied all the comments and criticisms raised by the reviewer. Here, let us reply to each of them.

“In this study, Sone et al. systematically investigate the interplay between nonlinear topological physics and bifurcation. As the triggering amplitude increases, bifurcation leads to chaotic behavior, ultimately resulting in the destruction of nonlinear topological boundary modes. Additionally, the nonlinear topological indices, as demonstrated using the plane-wave ansatz, may break down due to the emergence of chaos. The authors conduct comprehensive analytical and numerical research to explore the relationship between nonlinear topological phases and chaos. Remarkably, the analytical and numerical results align well. Although the study focuses on 1D systems, the findings are readily applicable to higher-dimensional systems.

I find the work to be well-written and intriguing, with the potential to merit publication in the journal Nature Communications. However, I have a couple of suggestions for the authors. If these questions are adequately addressed, I recommend considering this work for publication in Nature Communications.”

We are grateful to the reviewer for the positive appreciation of our results. We address each comment raised by the reviewer below.

“1. In Eq. (22), the authors specifically consider the special wavenumbers ($k_y = 0$) or (π) to demonstrate the applicability of their results in the 2D system. These wavenumbers serve as special reciprocal-space points where the nonlinear topological modes can be converted to chaotic modes as nonlinearity increases. However, it's essential to explore other wavenumbers as well (i.e., ($k_y \neq 0$) or (π)).

For different wavenumbers, the behavior of nonlinear boundary modes may vary. While the chaotic spatial profile arises for the special wavenumbers, it remains an open question whether similar chaotic behavior occurs for other wavenumbers. Thus, I suggest the authors exemplify the chaotic spatial modes for one or two wavenumbers other than 0 or π .”

Indeed, we can obtain the nonlinear dynamical system in the spatial direction and its chaos transition at different wavenumbers from $k_y = 0, \pi$. In Eq. (22), dispersion relations of edge modes are given by $E = \pm \sin k_y$. Assuming such wavenumber dependence of the nonlinear eigenvalue, we still obtain the same nonlinear dynamical system in the spatial direction as Eq. (7). Therefore, the chaos transition also occurs at the other wavenumbers. In general, it may be impossible to predict the nonlinear eigenvalues of edge modes. The extension to such edge modes with unpredictable eigenvalues is beyond the scope of this work and thus we remain it to a future work. We have added the sentence “Furthermore, even at $k_y \neq 0, \pi$, one can obtain a nonlinear dynamical system in the spatial direction similar to Eq. (7) (see also Methods).” in the second paragraph in the discussion section and the last paragraph in the section “Reduction of a higher-dimensional model into a one-dimensional model” in Methods to clarify these points (Summary of changes (19)).

“2. I had a look at the two references, one published in Phys. Rev. Lett. 132, 126601 (2024), “Bulk-Edge Correspondence for Nonlinear Eigenvalue Problems” by Isobe, et. al., the other paper is published by the current authors, namely “Nonlinearity-induced topological phase transition characterized by the nonlinear Chern number”, Nature Physics, by Sone et. al. It seems that these two papers have already indicated the nonlinear extension of bulk-boundary correspondence for $U(1)$ -symmetric systems, but now the authors indicate the breakdown of this bulk-boundary correspondence. Is there any inconsistency?”

There is no inconsistency between the present work and the previous works pointed out by the reviewer. In fact, we consider different types or strengths of nonlinearity from those in the previous papers. Specifically, the first paper (Phys. Rev. Lett. 132, 126601 (2024), “Bulk-Edge Correspondence for Nonlinear Eigenvalue Problems” by Isobe, et. al.) handled *linear dynamics* and thus the bulk-boundary correspondence is still established. Since they considered linear but higher-

order differential equations, the eigenvalue problems become nonlinear with respect to frequencies. In contrast, we here consider *nonlinear dynamics*, and thus the bulk-boundary correspondence can be broken. Therefore, the setup of our present work is fundamentally different from that of the previous work by Isobe et al. We have added the sentences “We note that a previous paper discussed the bulk-edge correspondence in nonlinear eigenvalue problems with respect to eigenfrequencies, which however describe linear dynamics of higher-order differential equations. In contrast, we here focus on the situation in which the dynamics itself is nonlinear.” in the second paragraph in the section “Setup” to clarify this (Summary of changes (20)).

On the other hand, the second paper (“Nonlinearity-induced topological phase transition characterized by the nonlinear Chern number”, Nature Physics, by Sone et. al.) indeed treated the same kind of nonlinearity as in the present work. In this previous work, we have shown that the notion of band topology can be extended to the mildly strong nonlinear regime, if we introduce the nonlinear topological number, by which the extended bulk-edge correspondence is restored. In contrast, in the truly strongly nonlinear regime, even the extended bulk-edge correspondence is broken down by bifurcation and chaos transition, which is the very main focus of the present paper. These observations are summarized in Figure 2 in a unified manner, and thus we have added the sentences “As shown in a previous paper for two-dimensional systems, the bulk-edge correspondence is valid from the weak to the mildly strong nonlinear regimes, by introducing the nonlinear extension of the topological number. However, in the truly strong nonlinear regime, the bulk-edge correspondence is broken down by the bifurcation and the subsequent chaos transition.” in its caption for further clarifications (Summary of changes (2)).

“3. In the original ArXiv version of the Nature Physics paper “Nonlinearity-induced topological phase transition characterized by the nonlinear Chern number”, there was a comprehensive discussion on nonlinearity that does not respect $U(1)$ symmetry. However, it appears that this discussion was removed in the published version. Given that many nonlinear interactions in nature violate $U(1)$ symmetry—such as those in mechanical, electrical, and biological systems—it is indeed crucial to address this aspect. I strongly recommend that the authors include a paragraph discussing nonlinear topological physics with interactions that break $U(1)$ symmetry in this current manuscript. This addition would enhance the relevance and completeness of their work.”

We thank the reviewer for this helpful comment. As the reviewer points out, the $U(1)$ symmetry is broken in some nonlinear systems including hydrodynamics and biological systems, which play important roles in nature. As inferred from interacting bosonic cases, the broken $U(1)$ symmetry may alter the topological classification of nonlinear systems. In our previous study [Ref. 49], we also found that the $U(1)$ symmetry is essential to assume that the bulk topology is described by the Bloch ansatz solutions. However, the extension to nonlinear systems without the $U(1)$ symmetry may still be possible by considering other ansatz states than the Bloch ansatz as in Ref. [64] (see also the reply to the fifth comment below). We have added the last paragraph in the section “More general cases” to discuss these points (Summary of changes (21)).

“4. Following the above-mentioned point #3, I recommend that the authors incorporate the reference titled ‘Nonlinear Topological Mechanics in Elliptically Geared Isostatic Metamaterials’ by Ma et al. (Physical Review Letters 131 (4), 046101, 2023). The inclusion of this paper is crucial for the following reason: When studying nonlinear topological mechanical modes in the Kane-Lubensky chain—similar to those investigated in Ref. [39] from the reference list—a chaotic mode should emerge. This behavior arises due to bifurcation in the Kane-Lubensky chain, leading to a chaotic spatial distribution of the mechanical mode. In contrast, the topological mechanical mode presented in ‘Physical Review Letters 131 (4), 046101 (2023)’ does not exhibit a chaotic boundary mode. The absence of chaos in this case results from the lack of bifurcation in the gear rotation angles for the nonlinear topological mechanical boundary mode. A comparative analysis between these two models would be insightful.”

We thank the reviewer for introducing a relevant paper, which we have cited in the revised manuscript as Ref.46. In response to this comment, we have done additional calculations on the model raised by the reviewer and clarified the reason for the absence of chaotic modes (see

Supplementary Note 6 for the detailed result). Specifically, we calculate the spatial dynamics of edge modes as in Fig. 3 in the present paper. The reason for the absence of the chaos transition is that the linear stability of the fixed point at zero does not change at any parameters. This situation is reminiscent of the nonlinear SSH models investigated in Refs. [36,41], where the fixed point that appears after the nonlinearity-induced topological transition is stable at any strength of nonlinearity. We have modified the sentence in the sixth paragraph in the section “Bifurcation and spatially chaotic zero modes” to “... for which the bulk-boundary correspondence has been confirmed under fairly strong nonlinearity (we show such wide-range stability of edge modes in a nonlinear topological mechanic in Supplementary Note 6).” and added new references mentioned by the reviewer [Ref. 46 and Supplementary Ref. 5] and Supplementary Note 6 to clarify these points (Summary of changes (22) and (S9)).

“5. Following the point #4 raised earlier, I recommend that the authors incorporate the reference titled ‘Topological Boundary Modes in Nonlinear Dynamics with Chiral Symmetry’ by Zhou (arXiv:2403.12480v2). In this work, the authors delve into nonlinear topological physics without $U(1)$ symmetry. It would be insightful to briefly discuss how the breakdown of bulk-boundary correspondence and the emergence of spatially chaotic modes manifest when $U(1)$ -symmetry-breaking nonlinearities are considered.”

We again thank the reviewer for introducing a relevant paper, which we have cited in the revised manuscript as Ref.70. The preprint (arXiv:2403.12480) has discussed the nonlinear Berry phase and its bulk-boundary correspondence under the existence of chiral (sublattice) symmetry but without assuming the $U(1)$ symmetry. The assumption of the modulated wave used in this related work can be useful to extend our result to nonlinear systems without the $U(1)$ symmetry. We emphasize that the breakdown of the bulk-boundary correspondence by the chaos transition also occurs without the $U(1)$ symmetry, while the nonlinear dynamical system in the spatial direction can be more complex than that with the $U(1)$ symmetry. We remain the concrete analysis to future studies. We have added the last paragraph in the section “More general cases” and a new reference mentioned by the reviewer [Ref. 64] to clarify this point (Summary of changes (21)).

“Overall, I find the analytical derivations to be reasonable, and the numerical results are convincing. Therefore, if these concerns are adequately addressed, I believe it would be appropriate for the manuscript to be published in the journal Nature Communications.”

Once again, we are very grateful to Reviewer #3 for the positive appreciation of our work and for the valuable comments. In particular, we have conducted an additional calculation on a model that exhibits no chaos transition. We do hope that the revised manuscript, together with our reply above, will meet with the reviewer’s approval.

Sincerely yours,

Kazuki Sone, Motohiko Ezawa, Zongping Gong, Taro Sawada, Nobuyuki Yoshioka, and Takahiro Sagawa

REVIEWERS' COMMENTS

Reviewer #1 (Remarks to the Author):

The authors have satisfyingly answered all the questions concerned. I think it can be accepted as it is.

Reviewer #2 (Remarks to the Author):

They authors have replied satisfactorily to all my remarks, and made appropriate changes and their manuscript. I therefore recommend publication of this work in the journal Nature Communication.

Reviewer #3 (Remarks to the Author):

The authors have thoroughly addressed my previous concerns. Their revisions and responses are more than satisfactory. I find their efforts commendable.

For instance, the authors' response to my Question #4 is particularly impressive. I initially sought a qualitative explanation of the relationship between linear stability and the absence of chaotic behavior. However, the authors exceeded my expectations by providing a comprehensive mathematical demonstration in the Supplementary Material.

Additionally, the response to my Question #3 is quite intriguing. Future research could explore chaotic modes even when $U(1)$ -symmetry is broken.

Overall, this response letter and revised manuscript exemplify the quality that referees appreciate. I strongly encourage the authors to maintain this revision style in their future works. Consequently, I highly recommend the publication of this current version in Nature Communications.